# The molecular basis, genetic control and pleiotropic effects of local gene co-expression

Diogo M. Ribeiro [1,2], Simone Rubinacci[1,2], Anna Ramisch [2,3,4], Robin J. Hofmeister [1,2], Emmanouil T. Dermitzakis [2,3,4] & Olivier Delaneau [1,2✉]

Nearby genes are often expressed as a group. Yet, the prevalence, molecular mechanisms and genetic control of local gene co-expression are far from being understood. Here, by leveraging gene expression measurements across 49 human tissues and hundreds of individuals, we find that local gene co-expression occurs in 13% to 53% of genes per tissue. By integrating various molecular assays (e.g. ChIP-seq and Hi-C), we estimate the ability of several mechanisms, such as enhancer-gene interactions, in distinguishing gene pairs that are co-expressed from those that are not. Notably, we identify 32,636 expression quantitative trait loci (eQTLs) which associate with co-expressed gene pairs and often overlap enhancer regions. Due to affecting several genes, these eQTLs are more often associated with multiple human traits than other eQTLs. Our study paves the way to comprehend trait pleiotropy and functional interpretation of QTL and GWAS findings. All local gene co-expression identified here is available through a public database (https://glcoex.unil.ch/).

[1] Department of Computational Biology, University of Lausanne, Lausanne, Switzerland. [2] Swiss Institute of Bioinformatics (SIB), Lausanne, Switzerland. [3] Department of Genetic Medicine and Development, University of Geneva, Geneva, Switzerland. [4] Institute of Genetics and Genomics in Geneva, University of Geneva, Geneva, Switzerland. ✉email: olivier.delaneau@unil.ch

Gene regulation is an essential component of cellular function, and its dysregulation often leads to disease[1,2]. The genetic makeup intrinsic to each person shapes their disease susceptibilities and response to treatment, making understanding the functional impact of genetic mutations one of the most pursued challenges in genetics research. Genome-wide association studies (GWAS) have associated tens of thousands of genetic variants with complex traits, but the functional link between them is unknown. In this context, expression quantitative trait loci (eQTL) analyses, which rely on studying the transcriptomic profiles of many individuals to reveal variants that modulate gene expression, are pivotal in bridging the functional gap between genotypes and organismal phenotypes[2–4]. Indeed, in combination with GWAS, eQTL studies have been crucial in identifying causal genes and tissues associated with complex traits and disease[5]. Yet, understanding a variant's molecular link to complex traits and disease is still a major challenge, given that most are found in the genome's non-coding regions, act only in specific tissues and may affect several genes[6,7].

A pertinent clue in the linking genome to phenome is the correlation of activity among regulatory elements and genes, which is key in understanding the shared genetic architecture between multiple complex traits and diseases as well as between all associated genes. In this respect, previous studies have found that gene order in eukaryotes is not random, and genes with similar expression profiles tend to be genomically linked[8]. Indeed, genes located near to each other (e.g. <1 Mb) often display concerted co-expression[9–11]. Local gene co-expression is more pronounced in the immediate vicinity of a gene (e.g. <100 kb) but it has also been shown that it extends further and occurs regardless of the strand, transcription orientation, shared functionality or tandem duplications[11,12]. The existence of structural and regulatory domains orchestrating the organised expression of nearby genes and mediating the genetic effects of regulatory variants on genes has been demonstrated[9], thus suggesting that regulatory element sharing and regulatory variant sharing may be key in controlling the co-expression of nearby genes. However, the molecular mechanisms, genetic control and tissue specificity of local gene co-expression are far from being fully understood. For instance, while genetic variants controlling expression have been found for almost all genes[2], the extent to which several genes may be controlled by the same genetic variants is yet unknown. Moreover, the full extent of local gene co-expression remains to be deeply assessed, particularly across tissues.

Here, we thoroughly investigate (i) the full genome-wide prevalence of local gene co-expression across human tissues, (ii) the molecular mechanisms that play a role in local gene co-expression, (iii) the regulation of local gene co-expression by genetic variants and (iv) their relevance on human disease and trait pleiotropy. For this, we develop a framework to detect local gene co-expression and associated regulatory variants using transcriptomic profiles across hundreds of genotyped individuals and 49 human tissues. To understand which molecular mechanisms play a role in local gene co-expression, we integrate various molecular features (e.g. chromatin contacts, enhancer–gene interactions) and estimate their ability to distinguish gene pairs that are co-expressed from those that are not. Furthermore, our study identifies thousands of eQTLs that are predicted to regulate multiple co-expressed genes through shared regulatory elements as well as associated with multiple societal-relevant diseases and traits. Our extensive search for local gene co-expression reveals novel links between genetic variation and gene (co-)expression, opening the way to comprehend the pleiotropy and disease comorbidity observed in humans and provide functional interpretation of QTL and GWAS findings.

## Results

**The prevalence of local gene co-expression across human tissues.** Previous studies have found extensive local gene co-expression throughout the genome[10,11], but no consensus approach to identify local gene co-expression exists. Moreover, to date, studies of local gene co-expression were only performed on a limited number of individuals and tissues[11] and although gene expression measurements are often confounded by both known and unknown confounders[13], these did not account for either.

Here, we developed a robust approach to generate genome-wide maps of local gene co-expression from cross-individual gene expression quantifications (e.g. a gene expression matrix from multiple RNA-seq experiments). For this, we exploited the principle that nearby genes exhibiting significant expression correlation between each other across individuals can be considered as co-expressed (Fig. 1). Briefly, for each gene, our method identifies all genes in a cis window of 1 Mb around the gene transcription start site (TSS) displaying substantial inter-individual expression correlation (Pearson correlation). Since the correlation value of truly correlated genes is unknown, we compare the observed values of each gene/cis-gene pair to expected correlation values under the null obtained by shuffling expression values, effectively ensuring that the number of nearby genes is accounted for (see the "Methods" section). In this way, high observed correlation values can be pinpointed and sets of bona fide co-expressed genes extracted while controlling for false discovery rate (FDR). Contrary to previous studies, this approach ensures that local gene co-expression can be identified regardless of gene order or transcriptional direction, which is particularly important given that gene regulation often occurs in a 3D manner. In addition, we extensively accounted for known (e.g. sex, subpopulation structure) and unknown confounding factors (using PCA for the Geuvadis dataset and PEER[14] for the GTEx dataset; see the "Methods" section).

We first applied our approach to a dataset of gene expression profiles from 358 lymphoblastoid cell line (LCL) RNA-seq samples from the Geuvadis project[15], composed of European individuals genotyped in the 1000 Genome project[16] (see the "Methods" section). We identified all correlated LCL-expressed gene pairs among 15,059 protein-coding and 1781 long intergenic non-coding RNA (lincRNA) genes (see the "Methods" section). At 1% FDR (corresponding to a minimum correlation coefficient of 0.16), we found 9384 significantly co-expressed gene pairs (COPs) within 1 MB of each other (8716 correlated positively and 668 correlated negatively), which correspond to 9030 distinct genes. A higher proportion of protein-coding genes (54.8%) is co-expressed compared to lincRNAs (43.5%), and as much as 53% COPs are formed between the nearest neighbours (Supplementary Fig. 1). Importantly, regardless of the FDR threshold used, we observe that COPs are well spread across all chromosomes (Supplementary Fig. 2), with the percentage of genes co-expressed falling between 46% and 62% across chromosomes (FDR 1%). Altogether, this indicates that local gene co-expression is highly prevalent in human LCLs.

When grouping COPs into network components—nodes being genes and edges being the co-expression between genes—we found 2545 unconnected groups of co-expressed genes (Fig. 2a). The majority of these groups are composed of only 2 or 3 genes (55.1% and 19.1% of the groups, respectively), yet, groups with more than 10 genes are also observed (3.5% of the groups). Several of these larger co-expressed gene groups can be readily assigned to known human gene clusters, such as the Hox A, B and C gene clusters[17], as well as a protocadherin cluster[18] and the histone H4 cluster[19] (Fig. 2a). To further understand how local

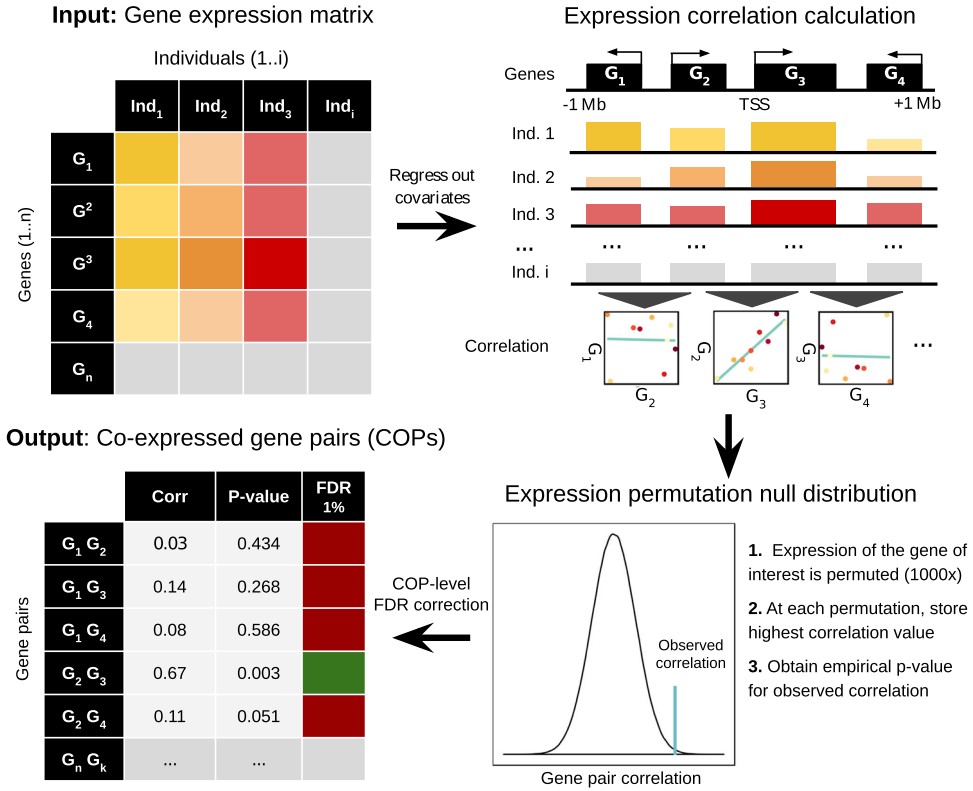

**Fig. 1 Local gene co-expression identification approach.** Inter-individual gene expression correlation is calculated for each gene pair whose TSSs are within 1 Mb of each other. For each gene, observed correlation values are compared to expected correlation values when shuffling expression values across individuals. Gene pairs passing the FDR threshold are considered COPs (see the "Methods" section).

gene co-expression is linked to shared functionality, we gathered data from several datasets of functional-relatedness (see the "Methods" section). When compared to all gene pairs tested, COPs are significantly enriched for (i) genes annotated with the same exact function, based on the Gene Ontology (GO)[20] (Fisher's exact test $p$-value $< 2.2e^{-308}$, odds ratio (OR) = 3.68), (ii) genes encoding proteins belonging to the same protein complex, based on CORUM[21] and hu.MAP[22] databases ($p$-value $= 2.1e^{-74}$, OR = 12.54), (iii) genes belonging to a same biological pathway, taken from KEGG[23] and Reactome[24] ($p$-value $= 2.8e^{-166}$, OR = 2.52; Supplementary Fig. 3). Substantial functional-relatedness in local gene co-expression could be due to local duplication of genes. In fact, we also found that COPs connect known paralogs more often than what is expected by chance (Fisher's exact test $p$-value $< 2.2e^{-308}$, OR = 11.20, Supplementary Fig. 3, see the "Methods" section). However, while these findings attest to the ability of our method in identifying known and expected groups of correlated genes, functional or evolutionary relatedness between gene pairs by no means explain all the local co-expression observed, as we could not find evidence of this link for the majority of COPs (71%, 6698 out of 9384 gene pairs).

To estimate the prevalence of local gene co-expression across human tissues, we applied our identification method to 49 human tissues from the GTEx v8 project[25] (see the "Methods" section). We identified 64,320 distinct COPs across tissues (FDR 1%, corresponding to a minimum correlation between 0.12 and 0.25, depending on the tissue). These amount to 18.3% of all gene pairs assessed for co-expression (350,564 gene pairs), which clearly demonstrate that local gene co-expression is highly prevalent. For individual tissues, the number of co-expressed genes ranged from 2347 (Brain Putamen basal ganglia; 13% genes tested) to 9062 (Thyroid; 50% genes) (Fig. 2b). The discrepancy in the numbers

of co-expressed genes may be due to distinct biological regulation of gene expression (e.g. 22,610 genes were found expressed in Testis against 15,051 in Whole Blood). However, we also found a strong correlation with the RNA-seq sample sizes available per tissue (Spearman rho = 0.75, $p$-value = $7e^{-10}$, Supplementary Fig. 4). Importantly, we observe that the patterns of gene co-expression sharing reflect well our expectations in terms of biological similarity across tissues. For instance, by measuring the percentage of COPs in a tissue that are also COPs in another tissue, we find that related tissues such as artery aorta and coronary artery or various brain subregions are concertedly grouped (Fig. 2c). Tissue grouping by gene expression of single genes had previously produced similar results[2], which indicates that the local gene co-expression observed here encompasses relevant gene pair links and biological meaning. Our results about local gene co-expression across tissues, their correlation values, statistical significance and genomic information are all readily available in the LoCOP public database (http://glcoex.unil.ch).

**Molecular features associated with local gene co-expression.** Gene expression is known to be regulated by several genomic elements such as promoters, enhancers and insulators[26,27]. Here, we assess whether regulatory elements and mechanisms (termed here as 'molecular features') known to be associated with gene expression regulation can explain the observed inter-individual local gene co-expression. For this purpose, we train logistic regression models (on 80% of the gene pairs) and measure the area under the ROC curve (AUC) on predictions of a test set (20% of the gene pairs) to compare how well local gene co-expressed genes pairs (i.e. positive cases) can be distinguished from non-co-expressed pairs (i.e. negative cases) for each assessed molecular feature (see the "Methods" section). For instance, as reported in previous studies[11,26], genes that are co-expressed

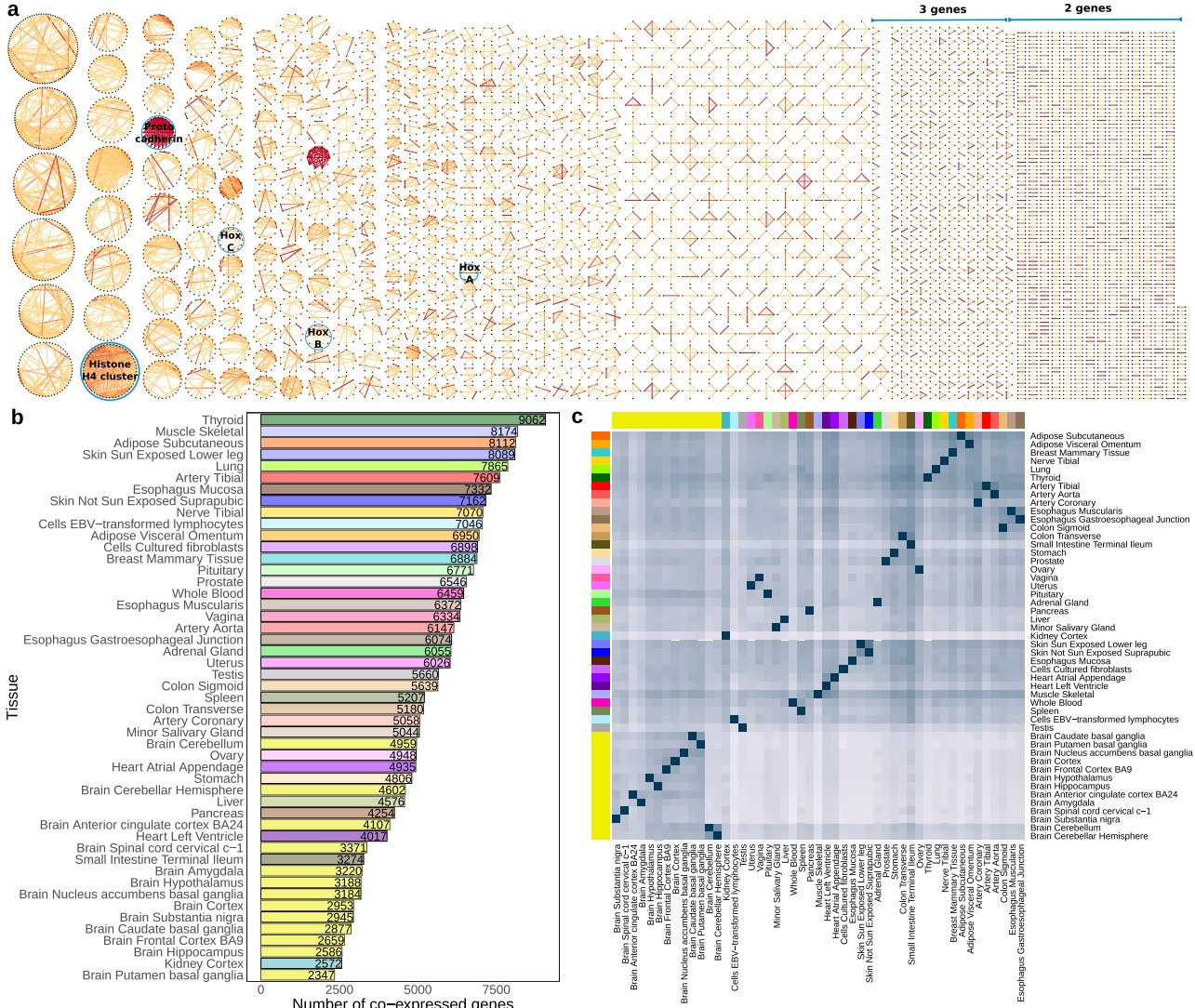

**Fig. 2 Local gene co-expression discovery across tissues. a** Grouping of Geuvadis LCL COP discovery into 2545 network components (genes as black nodes, correlation as edges), edge colour is mapped to correlation strength and several known gene clusters are highlighted. **b** Number of COPs identified in each GTEx tissue. **c** Clustering of COP sharing across GTEx tissues. For each pair of tissues (tissue A, tissue B), the percentage of COPs of tissue A that are also COPs in tissue B is calculated and used for clustering (hierarchical clustering). Higher blue colour intensity corresponds to a higher percentage of COPs shared.

locally are more likely to be found in close proximity. Indeed, even though we searched for co-expression of genes within 1 Mb of each other, we found that most COPs are found in relatively closer distances, with 71.7% COPs found within 200 kb from each other (Fig. 3a; median distance = 78,543 bp. Geuvadis LCLs). Predicting local gene co-expression through genomic distance alone provides an AUC of 0.81 (Supplementary Fig. 5). While clearly important, distance alone does not fully determine local gene co-expression, as only a fraction of nearby gene pairs are found co-expressed (e.g. 11.8% of gene pairs apart for <200 kb are co-expressed; 17.1% for 100 kb; 34.4% for 10 kb). This hints at the presence of fine-detailed regulatory mechanisms that determine which gene pairs are co-expressed and the presence of genomic structure (e.g. chromatin boundaries) which effectively separate in space and time the expression of nearby genes.

Given that many molecular features—such as DNA contacts and gene–enhancer interactions—are dependent on genomic distance, we compare COP's molecular features to those observed in a set of non-co-expressed gene pairs (non-COPs) that very closely matches the distance distribution observed in COPs

(Fig. 3a; see the "Methods" section). This ensures that the following results are independent of distance (i.e. distance has an AUC of 0.5). Importantly, since paralog genes may display co-expression due to particular circumstances, such as the use of the same or a duplicated regulatory element, all gene pairs composed of paralog genes were excluded from this and all subsequent analyses to exclude this potential bias (see the "Methods" section). Moreover, for simplicity, in all analyses, we focused on COPs with positive expression correlation, for which we have approximately an order of magnitude more COPs than for negative correlation. Results on negatively correlated COPs are summarised in Supplementary Note 1. After these filters, 6668 Geuvadis COPs and 40,999 distinct GTEx COPs (number of COPs per tissue on Supplementary Fig. 6) were kept for analysis.

We first analysed COPs derived from Geuvadis LCLs, for which there is a wealth of molecular annotations. When measuring the presence of total and inverted CTCF-binding sites between gene TSSs (suggestive of insulation) we found them to be clearly depleted in COPs when compared to distance-matched non-COPs, in particular when distance increases (Supplementary

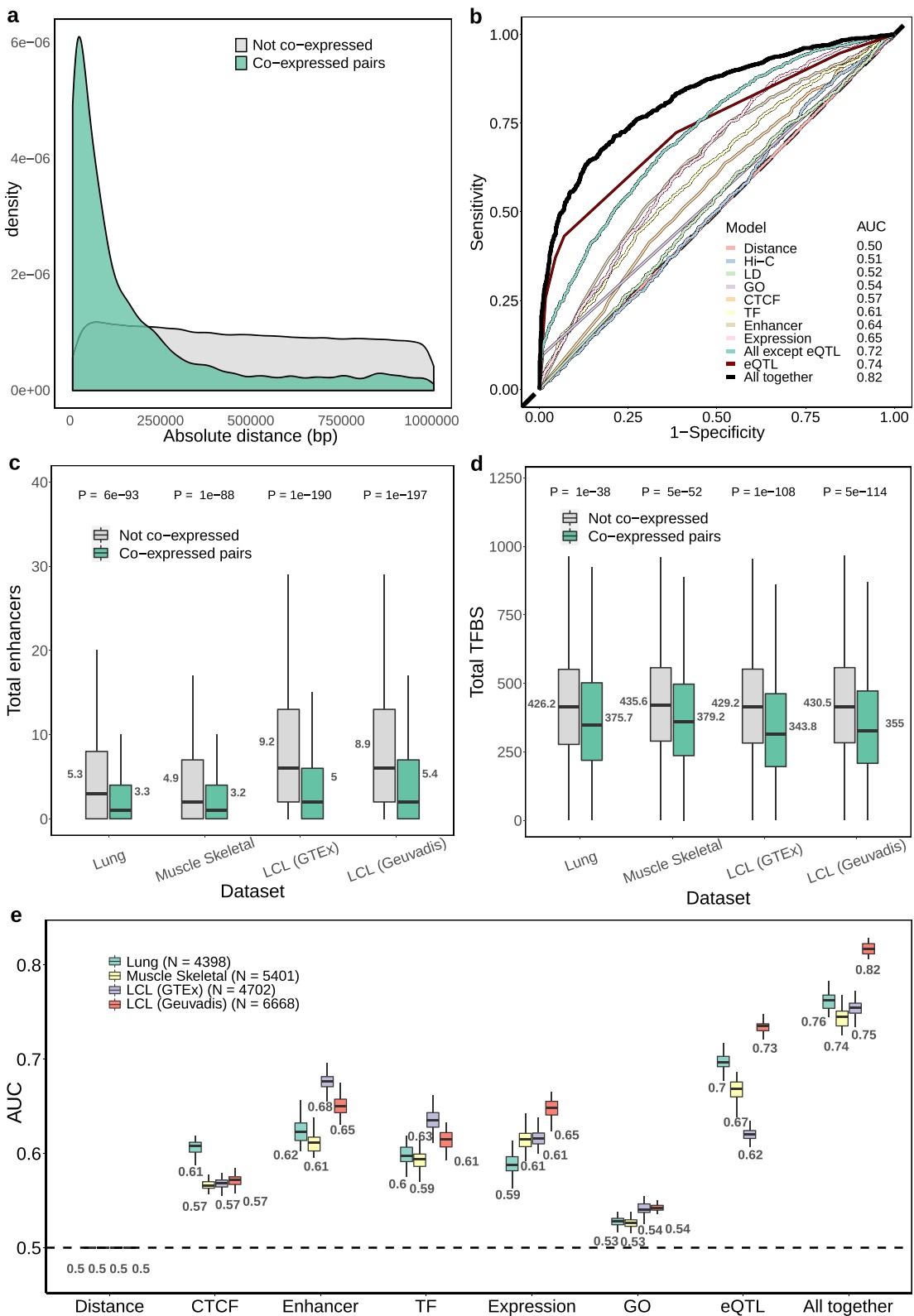

**Fig. 3 Molecular features of COPs. a** Distribution of the absolute distance between TSS for Geuvadis LCL COPs and non-COPs. Gene pairs before applying paralog and positive correlation filters. **b** Receiver operating characteristic (ROC) curve of predicting Geuvadis LCLs COPs for several molecular features (logistic regression; $N = 6668$ for COPs and for non-COPs; see the "Methods" section for molecular feature descriptions). **c** and **d** boxplots of total enhancers and total transcription factor binding sites (TFBS), respectively, between COPs and non-COPs across four datasets: Geuvadis LCLs ($N = 6668$), GTEx LCLs ($N = 4702$), Lung ($N = 4398$) and Muscle Skeletal ($N = 5401$). Values next to the boxplots represent the mean. $P$-values were obtained from two-tailed Wilcoxon signed-rank tests. **e** Boxplots of the AUC values obtained for each molecular feature and dataset across the 50 training-test set randomisations. Values below each boxplot represent the mean AUC. For each boxplot, the length of the box corresponds to the interquartile range (IQR) with the centre line corresponding to the median, the upper and lower whiskers represent the largest or lowest value no further than 1.5 * IQR from the third and first quartile, respectively.

Fig. 7; see the "Methods" section). This makes CTCF-binding site presence reasonably predictive of local gene co-expression (mean AUC = 0.57 for all COPs, 0.63 for those >200 kb apart; Fig. 3b; Supplementary Fig. 8), which indicates possible constraints in the chromatin structure in which some gene pairs may be hindered from interacting. Looking more closely at chromatin interactions, we found that chromatin contacts between gene TSSs (Hi-C data, 5 kb resolution; see the "Methods" section) are higher in COPs than non-COPs when these are separated by more than 200 kb (mean AUC = 0.59), whereas at closer distances such contacts occur with similar intensities in COPs and non-COPs (mean AUC = 0.50; Supplementary Figs. 7 and 8). These results indicate that chromatin structure may play a role in gene co-expression, yet, these do not seem determinant in pinpointing which local gene pairs are co-expressed or not.

Next, we assessed whether enhancer–gene interactions and transcription factor (TF)-binding explain local gene co-expression. Interestingly, COPs were found associated with lower numbers of enhancers than non-COPs (LCL-active enhancers; Fig. 3c). Moreover, a similar pattern was observed when analysing TF binding on the 50 kb flanking region of gene TSSs (see the "Methods" section), where again we found that COPs have fewer TF-binding sites (TFBS) than non-COPs (Fig. 3d). These results were readily replicated in LCLs from the GTEx project (Fig. 3c, d), as well as in two other GTEx tissues for which tissue-specific enhancer datasets were available—Skeletal Muscle and Lung (mean AUCs between 0.59 and 0.68, Fig. 3c–e). Interestingly, when we split COPs into four categories based on their tissue prevalence across 49 tissues (Supplementary Fig. 9; see the "Methods" section), we observed a clear trend in which conserved COPs generally have higher AUCs across molecular features than tissue-specific or unique COPs (Supplementary Fig. 10). Indeed, the AUCs for conserved COPs are consistently higher than tissue-specific COPs for each molecular feature, including a more pronounced depletion of TFBSs and interacting enhancers (Supplementary Fig. 10). Importantly, these results are consistent for COPs and annotations from Muscle Skeletal, Lung and LCL (Supplementary Figs. 10–12). A hypothesis for this finding is that there is a driving force for COPs to reduce the total number of expression regulators, i.e. a pressure to keep regulatory machinery relatively simple in order to achieve similar levels of expression between genes. Indeed, another feature of COPs is that the expression level and variation between the two genes in the pair tend to be more similar than when compared to non-COPs (mean AUC between 0.59 and 0.65, Fig. 3e, Supplementary Fig. 13). Importantly, this is observed even when matching the expression levels between COPs and non-COPs (Supplementary Fig. 13).

We confirmed that the molecular features observed for COPs are largely independent of the linkage disequilibrium (LD) between the genes' promoters (mean AUC = 0.52, Geuvadis LCLs, Fig. 3b; see the "Methods" section). Indeed, the genetic linkage of COPs and non-COPs does not substantially differ, as measured by their TSS centimorgan distance and presence in the same LD block (Supplementary Fig. 14). Moreover, since paralog genes were excluded from analysis, the sharing of function between gene pairs also had low AUCs (0.53–0.54 for Lung, Muscle Skeletal and LCL, Fig. 3e), as measured by the two genes in the pair sharing the same exact biological process (BP) GO term (see the "Methods" section). Interestingly, when combining all the molecular features assessed in Geuvadis LCLs together into the same regression model we obtain a global mean AUC of 0.72 (Fig. 3b). This AUC is higher than any individual molecular feature, indicating that each feature explains local gene co-expression in a complementary way. Indeed, except for Hi-C contacts and redundant metrics, each distinct molecular feature

significantly explains some of the variations in local gene co-expression independently of the other features (Supplementary Table 1) and no combination of two features reaches AUC levels above 0.69 (the highest being total enhancers in combination with expression level similarity; Supplementary Fig. 15). Albeit reasonable, a global AUC of 0.72 also indicates that the molecular features or the datasets used here are yet far from explaining all the observed local gene co-expression. Supplementary Note 2 compares a subset of the assessed molecular features between cis-COPs and trans-COPs.

One way to assess the effect of known and unknown molecular features and mechanisms implicated in gene regulation is to exploit the natural genetic variation observed in human populations (e.g. eQTLs). To assess how well genetic variation can provide information about gene co-expression, we identified eQTLs in cis (±1 Mb) for all expressed human genes, i.e. discovered associations between genetic variants and gene expression levels for all genes across all tissues (FDR 5%; see the "Methods" section). To discover putative cases of co-regulation of gene pairs by the same genetic variant(s), for each gene pair, we tested whether the lead eQTL of each gene (i.e. the genetic variant most strongly associated with the gene, if any) is also associated to the other gene in the pair (nominal $p$-value < 0.05; see the "Methods" section). For simplicity, we name such associations of a variant to gene pairs as 'eQTL sharing'. Through this approach, we found that as many as 41.9% of all COPs display eQTL sharing in Geuvadis LCLs (358 genotyped individuals), compared to only 5.9% in distance-matched non-COPs (Supplementary Fig. 16). Given that eQTL discovery power is largely dependent on sample size and only 46.2% genes (7805 out of 16,906 expressed genes) are genes (i.e. significantly associated with a cis variant) in Geuvadis LCLs, the high number of co-regulated COPs is striking. In fact, if only gene pairs composed of eGenes are considered (i.e. genes significantly associated with at least one eQTL), the percentage of COPs in eQTL sharing increases to 85.4%, compared to 20.0% in non-COPs (Supplementary Fig. 16). Moreover, we observed that eQTL sharing occurs more often for local co-expressed genes than genes that are co-expressed in trans at similar correlation values (trans-COPs, see the "Methods" section). Indeed, only 8.9% of trans-COPs were found in eQTL sharing (compared to 41.9% cis-COPs; Supplementary Fig. 16), suggesting that most eQTL sharing found for cis-COPs may represent co-regulation of multiple genes by the same variant rather than simple transient correlation between expression values and genotypes.

In the regression model, eQTL sharing leads to mean AUCs between 0.62 and 0.73, depending on the tissue assessed (Fig. 3e). Importantly, by combining eQTL sharing and all previously assessed molecular features, we increase the global mean AUC to between 0.74 (Muscle Skeletal) and 0.82 (Geuvadis LCL, Fig. 3b, e). This increase in the global model AUC indicates that by measuring eQTL sharing we are able to capture gene regulation that was not yet assessed with the other molecular features. Overall, these results highlight the complexity and diversity of mechanisms involved in the regulation of gene co-expression and indicate a high level of control of co-expression regardless of the genomic distance between genes.

**Genetic regulation of local co-expressed genes**. The integration of QTL analysis and functional annotations provides insights into the molecular mechanisms of transcriptional regulation and their phenotypic consequences. We thus next investigated whether shared eQTLs are more likely to fall in regulatory regions of the human genome than lead eQTLs that associate with a single gene. First, as expected, we found that all Geuvadis LCL lead eQTLs

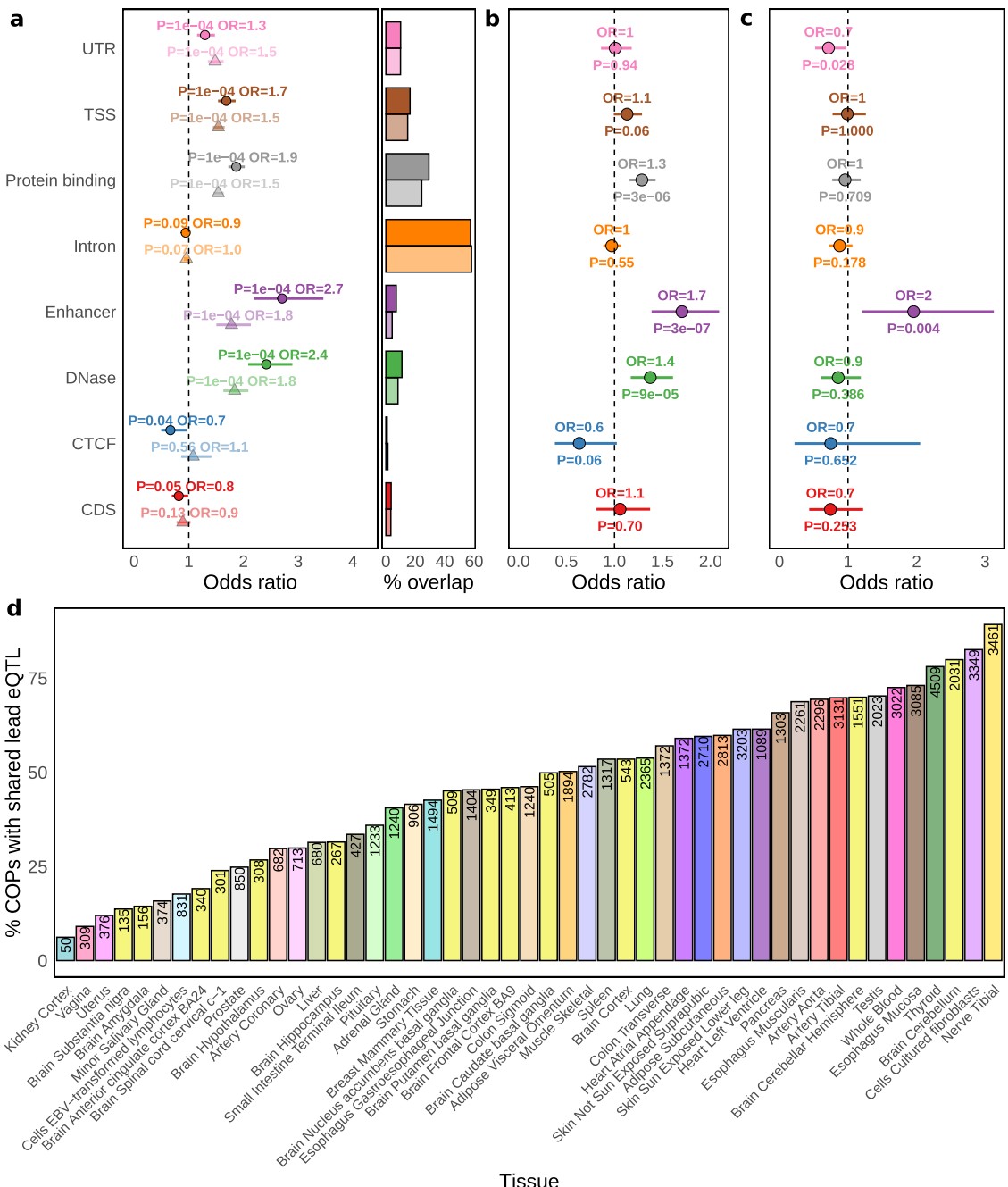

**Fig. 4 Discovery and functional enrichments of shared eQTLs. a** Overlap enrichment of Geuvadis LCL shared lead eQTLs (solid colour, round points) and other lead eQTLs (pale colour, triangles) in Encode LCL functional annotations and Gencode gene body categories (see the "Methods" section). Odds ratios are calculated based on the observed versus expected overlap (10,000 QTLtools Fenrich permutations; see the "Methods" section) between eQTLs and each functional annotation, through two-sided Fisher's exact tests (no multiple test adjustment). Error bars are 95% confidence intervals. The right part of the plot denotes the percentage of overlap between eQTLs and each functional annotation. **b**, **c** Two-sided Fisher's exact test odds ratio and p-value for the enrichment of shared lead eQTLs in each functional annotation compared to other lead eQTLs, for eQTLs in Geuvadis LCLs and GTEx LCLs, respectively. Error bars are 95% confidence intervals. **d** Percentage of COPs with a shared lead eQTL per GTEx tissue. Numbers inside bars denote the number of COPs with eQTL sharing.

(shared or not) are more likely than expected by chance to fall within gene body regions as well as various LCL-specific functional regions from Encode[28] such as enhancers (OR = 1.8–2.7, p-value = 1e$^{-4}$), DNAse sensitive regions (OR 1.8–2.4, p-value = 1e$^{-4}$) and protein binding regions (OR = 1.5–1.9, p-value = 1e$^{-4}$, Fig. 4a; see the "Methods" section). However, we found that shared eQTLs in LCL are more highly enriched in these three functional annotations than other lead eQTLs, particularly for

enhancers (Fisher's Exact test OR = 1.7, p-value = 3e$^{-7}$, Fig. 4b). Remarkably, an enrichment for shared eQTLs overlapping enhancers more than other eQTLs can also be observed for GTEx LCLs (Fisher's Exact test OR = 1.96, p-value = 4e$^{-3}$, Fig. 4c), as well as for functional annotations from the Roadmap Epigenomics project[28,29] for LCL, Muscle and Lung tissues (Fisher's Exact test OR = 1.3–1.4, Supplementary Figs. 17–20). Crucially, discovering genetic variants affecting regulatory regions such as

**Table 1 Number of lead eQTLs associated with one or more traits.**

| Lead eQTL category | Total # eQTLs | Associated to a trait | Associated to >1 trait |
|---|---|---|---|
| All lead eQTLs | 145,327 | 14,934 (10.3%) | 3921 (26.3% of previous) |
| Shared lead eQTLs | 32,636 | 4301 (13.2%) | 1274 (29.6% of previous) |
| Other lead eQTLs | 112,691 | 10,633 (9.4%) | 2647 (24.9% of previous) |

enhancers, which are known to interact with several genes, can provide a mechanistic explanation of how such variants regulate multiple genes. To further evidence that COPs are co-regulated by the same genetic variants, we performed colocalization analysis between the eQTL signals of the two genes in the pair using COLOC[30] (see the "Methods" section). As expected, a large portion of COPs in eQTL sharing show evidence of common eQTL signals (45.6% COPs, coloc H4 posterior probability (PP4) > 0.5), whereas COPs that are not in eQTL sharing show very low evidence of common eQTL signals (0.9% COPs with PP4 > 0.5, Supplementary Fig. 21). Importantly, COPs (in eQTL sharing or not) also evidence more common eQTL signals than non-COPs, even when considering only pairs of genes in both categories (PP4 > 0.5 in 42.4% of COPs compared to 5.4% for non-COPs, Supplementary Fig. 21). In addition, we find that shared eQTLs with PP4 > 0.5 display stronger functional enrichments than those with PP4 < 0.5 (Supplementary Fig. 21). Yet, even shared eQTLs with PP4 < 0.5 are enriched for various annotations, including enhancers (OR = 2.2, $p$-value = 1e$^{-4}$).

To study the sharing of genetic regulation across all 49 GTEx tissues, we extended our approach to identifying all lead eQTLs affecting the expression of COPs to all tissues (5% FDR, see the "Methods" section). We found that 63% (85,460 out of 135,662) COP–tissue pairs have a lead eQTL that is significantly associated with both genes, illustrating the widespread sharing of genetic effects through local co-expressed genes. Per tissue, between 6.2% and 89.3% (median = 46.2%) COPs have a shared lead eQTL (Fig. 4d), the large spread between tissues being largely due to sample size differences (Spearman rho = 0.78, $p$-value = 3.5e$^{-11}$, Supplementary Fig. 22). On average, a COP in eQTL sharing is associated with the same lead eQTL in 21.8% of the tissues where present (considering COPs in >5 tissues, Supplementary Fig. 23). As a whole, these results indicate that the co-regulation of multiple genes by single genetic variants is widespread and often made through shared regulatory elements.

**Shared gene regulation and trait pleiotropy.** Recent GWAS and PheWAS studies have uncovered extensive pleiotropy of complex trait associations, finding that a high proportion of variants are associated with multiple traits[31]. A plausible explanation for this is the association of variants to multiple genes. Indeed, cis-eQTLs that affect the expression of multiple genes have been recently shown to have high complex trait pleiotropy[25]. Given the abundance of genetic variation co-regulating multiple genes across tissues found in our study (shared eQTLs), we sought to investigate how pleiotropy can be driven by gene co-regulation. For this, we first collected GWAS summary statistics for 35 traits from the PanUK Biobank (Pan-ancestry genetic analysis of the UK Biobank) including binary and continuous data types, from basic anthropometric measurements to frequently studied clinical and complex traits with more than 15,000 cases (Supplementary Table 2; see the "Methods" section). Association summary statistics from the PanUK Biobank are provided for the same set of 28,987,534 variants across traits and these include the vast majority (96.8%) of all distinct lead eQTLs found across tissues. Then, for each trait, we only retained genome-wide significant

associations ($p$-value < 5e$^{-8}$), resulting in 373,206 variants, of which 109,167 (29.3%) are associated with more than one of the 35 traits. Considering the 32,636 unique lead eQTLs across tissues associated with COPs (eQTL sharing), we found that 4301 (13.2%) are associated with at least one trait, with 1274 (29.6% of those) associated with more than one trait (Table 1). This compares to 10,633 (9.4%) of other lead eQTLs (not shared) being associated with at least one trait, with 2647 (24.9% of those) associated with more than one trait. Shared eQTLs are thus significantly more likely to be associated with at least one trait (Fisher's Exact test OR = 1.5, $p$-value < 2e$^{-16}$), as well as with more than one trait (OR = 1.7, $p$-value < 2e$^{-16}$). Indeed, out of all lead eQTLs associated with traits, those that are shared are more pleiotropic than other lead eQTLs (mean 1.5 for shared, 1.38 for other lead eQTLs, two-sided Wilcoxon rank-sum test $p$-value = 3e$^{-11}$, Fig. 5a). We confirmed that shared eQTLs have increased pleiotropy independently of their increased presence in multiple tissues (two-way ANOVA $p$-value = 1.4e−11). Indeed, the presence of the same lead eQTL across multiple tissues, previously found to have higher pleiotropy than tissue-specific eQTLs[25], results only in a mild increase of pleiotropy in our dataset (mean 1.45 for multiple tissues, 1.4 for tissue-specific, two-sided Wilcoxon rank-sum test $p$-value = 0.06, Supplementary Fig. 24). This suggests that pleiotropy can arise due to the co-expression and co-regulation of nearby genes, for example, through the effect of several genes implicated in diverse physiological processes.

Next, we investigated whether shared lead eQTLs are strongly associated with the 35 traits analysed here, by comparing the probability distributions of all variants prior to any $p$-value filtering. First, as found in the previous studies[5,32], we observed that lead eQTLs (shared or other) are more likely to display stronger associations (i.e. lower association $p$-value) than other variants, as observed in quantile-quantile plots (Q–Q plots) and measured by the genomic inflation factor ($\lambda$ metric, Fig. 5b, Supplementary Fig. 25). Notably, we found that shared lead eQTLs often have stronger associations than other lead eQTLs, as it can be observed in known tissue-trait pairs, such as hypothyroidism-thyroid ($\lambda = 1.64$ for shared, $\lambda = 1.34$ for other eQTLs) and systolic blood pressure—artery aorta ($\lambda = 2.83$ for shared, $\lambda = 2.55$ for other eQTLs) (Fig. 5b). Moreover, considering all shared and other lead eQTLs across the 49 GTEx tissues, we found stronger association $p$-values (higher $\lambda$) for shared lead eQTLs across 34 out of the 35 traits assessed, which is a remarkably consistent result (Supplementary Fig. 25). Together, these findings suggest that eQTLs affecting more than one gene play an important role in (multiple) complex traits and disease, and can help reveal novel links between traits that may explain disease and trait comorbidity.

Through our analysis, we identified a total of 1274 unique variants (in 3201 eQTL–COP–tissue combinations) that affect multiple genes as well as multiple traits (Supplementary Data 1). Of these variants, 219 fall in promoters, and 26 in enhancer regions, a higher overlap proportion than found for other lead eQTLs that are also pleiotropic but do not affect multiple genes (Fisher's exact test $p$-value = 1e$^{-5}$ and 0.05, respectively, Supplementary Fig. 26, Ensembl v101 Regulatory Build[33]). This overlap to functional regions denotes plausible mechanistic

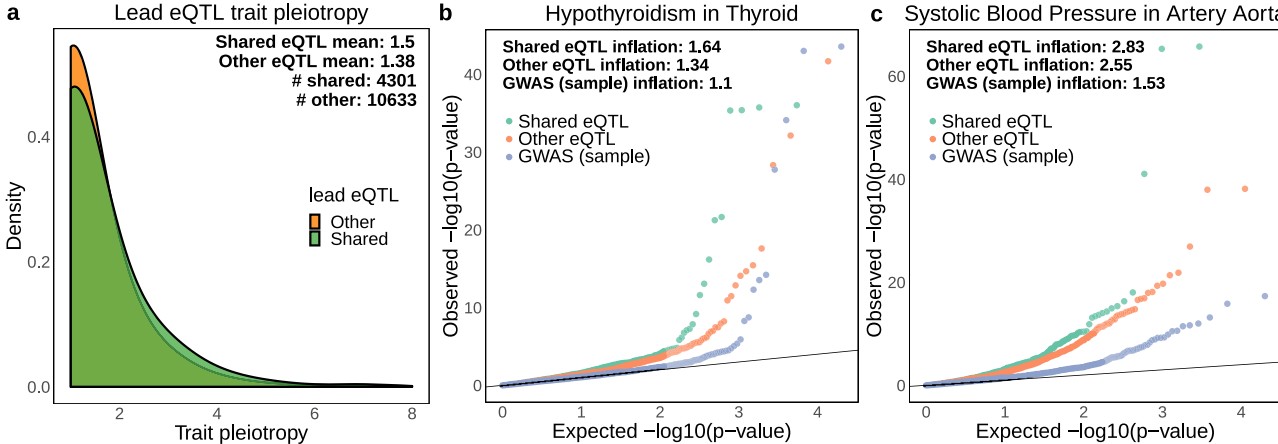

**Fig. 5 Comparison of trait pleiotropy and association *p*-values between shared and other lead eQTLs. a** Distribution of trait pleiotropy (association *p*-value < 5e−8) across 35 UKBB GWAS traits. If a lead eQTL is shared in at least one tissue (even if not in other tissues) it is considered as "shared". The remaining lead eQTLs compose the "other" category. Only lead eQTLs associated with at least one trait were considered; **b, c** Q–Q plots of association *p*-values (−log10 scale) for known tissue-trait matched pairs (hypothyroidism in thyroid tissue and systolic blood pressure in artery aorta), comparing shared (green) and other (orange) lead eQTLs. GWAS (blue) is a sample of 10,000 variants (randomly and independently picked for each trait) shown only for comparison purposes.

explanations linking genetic variation to the expression of several genes which in turn affect multiple traits. Overall, our approach allows us to gather novel information into the genetic architecture of disease loci, and its exploration can open the way to comprehend trait pleiotropy and comorbidity, aiding the interpretation of QTL and GWAS findings.

## Discussion

Prevalent genome-wide local gene co-expression had been previously reported in humans, but limited to a few tissues or cell types[10,11]. In this study, we confirm that local gene co-expression is highly prevalent and we provide the community with a database containing an extensive catalogue of local gene co-expression and shared eQTLs across 49 human tissues. Previous studies reported a genome-wide level of 37.4% co-expressed genes (out of 20,502 genes assessed) in 100 normal breast tissue samples[10]. When correcting for multiple testing and accounting for technical variability, as we attempted in our new method, we found a similar proportion of co-expressed genes (38.1%), even though a larger dataset comprising 396 GTEx breast mammary tissue samples was used. Interestingly, we found that COP genes are enriched in belonging to the same biological pathway (OR = 2.6, Supplementary Fig. 3), BP GO term annotation (OR = 3.9) and protein complex (OR = 12.5). The particularly high enrichment in COPs encoding proteins belonging to the same protein complex could indicate that gene co-regulation may aid the maintenance of stoichiometry in protein complexes. Our finding that COPs have more similar expression levels and variation compared to non-COPs corroborates this notion.

In this study we assessed the relevance of several molecular features for local gene co-expression, replicating results in various tissues and datasets. As expected from the perceived complexity of gene regulation, we found that several factors contributed to local gene co-expression and provided novel insights into cis-regulation. Of particular relevance was the finding that low regulatory complexity, e.g. a low number of nearby TFBS and enhancers, coincides with high gene co-expression, particularly in COPs present in many tissues. In agreement with our findings, a recent study identifying cohesin chromatin loops also evidenced the usage of a simpler circuitry in constitutive genes which have a steady level of expression, whereas more extensive regulatory architectures were found in dosage-sensitive genes[34].

A limitation of our approach rests in our inability to disentangle whether the observed molecular features (e.g. enhancer-gene interactions, CTCF binding) actually cause the observed gene co-expression or whether they are its consequence. To discriminate these, molecular experiments would have to be performed (e.g. massively parallel reporter assays and CRISPR/Cas9-based technologies[27,35]). Another feature that is hard to discriminate between cause or consequence, is the finding that genes in a COP often have similar expression levels as well as reduced expression variation between them. On one hand, selection for reduced expression noise has been previously proposed as a cause for the grouping of genes in the genome, particularly for essential genes[36]. On the other hand, local gene co-expression can simply be a consequence of stochastic chromatin fluctuations (i.e. similar expression patterns can be a consequence of genes being close together). Indeed, Kustatscher et al. have found evidence for both cases, although most local gene co-expression was shown to be buffered at the protein level[37].

Our approach is affected by the amount and quality of molecular data available for distinct tissues. This could explain some of the differences in AUCs observed, such as lower AUCs for enhancer features for Lung and Muscle skeletal (AUC = 0.57), as only 43,973 and 39,708 predicted enhancer–gene interactions were used for Lung and Muscle skeletal, respectively, compared to 78,796 for LCLs (AUC = 0.61 for GTEx and 0.64 for Geuvadis LCLs)[38]. Indeed, given the difficulty in obtaining complete tissue-specific regulatory maps (e.g. all enhancer–gene interactions relevant in a certain tissue) and the added complexity of regulatory mechanisms (e.g. we did not include CpG islands and promoter types in our study), we did not fully account for all possible regulatory mechanisms in this manner. Instead, we reasoned that identifying natural genetic variation and associating it with the expression of several genes (shared eQTLs) would allow us to map genomic regions pertaining to both known (e.g. enhancers) and unknown features of gene co-regulation. Indeed, the AUCs for eQTL sharing were shown to be higher than for any other metric assessed, which indicates the feasibility and benefit of this approach to study the regulation of local gene co-expression.

While our approach to find variants affecting multiple genes may be overly simplistic (e.g. only lead eQTLs are used and causality is not assured) and limited by sample size, we have

nevertheless found extensive evidence of joint control of co-expressed genes through the same genetic variant. Indeed, we found that shared eQTLs (i) often overlap enhancers, (ii) are more strongly associated with GWAS traits and (iii) are more likely to affect more than one trait. Recent studies also found that eQTLs that affect the expression of multiple genes display high complex trait pleiotropy[25,39]. Our findings provide further insights into the molecular underpinnings of pleiotropy: variants that affect the expression of multiple genes by overlapping regulatory elements display a higher degree of complex trait pleiotropy. This indicates that the pleiotropy reported in previous studies[31] may emerge from cis-regulation, besides the possibility of it arising from multiple functions of genes, particularly under distinct tissues and cellular contexts. Moreover, describing COPs and their co-regulation can aid in establishing the direction of causality between traits that were previously found to be comorbid. Characterising how variants are functionally linked to disease is still a challenge but approaches such as ours provide an additional layer of interpretation, by linking groups of genes regulated together and describing how regulatory signals may be propagated. As current colocalization methods are only focused on the sharing of GWAS and eQTL signals for single genes, the development of methods dedicated to pinpointing causal variants co-regulating multiple genes and/or other molecular phenotypes may prove an asset for future research.

## Methods

**Geuvadis consortium gene expression dataset**. Geuvadis consortium[15] BAM files previously mapped to GRCh37 for RNA-seq experiments on LCL were used for COP identification. Only a subset of 358 European (EUR) individuals from the Geuvadis study also present in the 1000 Genomes project phase were considered. Data was downloaded from the EBI ArrayExpress (accession code E-GEUV-1). Gene expression was quantified for all protein-coding and lincRNAs geness annotated in GENCODE v19[40] using QTLtools[41] v1.1 *quan* function with default parameters. Genes within or around the MHC complex region (chr6:29500000–33600000) and in non-autosomal chromosomes (X and Y), as well as pseudoautosomal regions (PAR1, PAR2) or unassembled regions, were removed. Moreover, genes with no expression measurements across most individuals (≥50% of the individuals with RPKM = 0) were also excluded. To account for confounding factors, the following covariates were regressed out: (i) sample sex (as defined by the 1000 Genomes project), (ii) ancestry (three first PCA principal components (PC) computed using QTLtools *pca* function on genotype data that has been trimmed for one variant with minor allele frequency (MAF) > 5% every 50 kb) and (iii) unknown technical/experimental variables based on the first 50 PCA PCs computed using QTLtools *pca* function on the gene expression matrix. The 50 PCs were determined as the number of PCs that maximises the number of eGenes discovered. Importantly, residualising gene expression by several PC, a practice commonly used in eQTL mapping[2] to improve discovery power, is expected to reduce trans gene co-expression effects, leading to increased power in detecting weaker effects of local gene co-expression. Finally, the expression quantifications across individuals were normalised to match a normal distribution $N(0,1)$.

**GTEx project gene expression dataset**. Gene expression quantifications (TPM values) from RNA-seq experiments across 49 tissues (for which genotype data is also available for ≥70 individuals) processed and provided by the GTEx project v8[25] were used for COP identification. Data was downloaded from dbGaP (accession: phs000424.v8.p2). The provided quantifications had been mapped to Gencode v26[40] gene annotations on hg38 and normalised by TMM between samples (as implemented in edgeR), and inverse normal transform across samples. Moreover, only genes passing an expression threshold of >0.1 TPM in ≥20% samples and ≥6 reads in ≥20% samples had been retained. As done for the Geuvadis dataset, only protein-coding and lincRNA genes were considered in this study, and genes in the MHC complex region and in non-autosomal or pseudoautosomal regions were removed. Covariates provided by GTEx v8 for each tissue were regressed out of each expression matrix to account for potential confounding factors. These included 15–60 Peer factors (depending on tissue sample size), 5 Genotype PCA PCs as well as information about the sequencing platform, PCR usage, and the sex of the samples, all provided by GTEx.

**Identification of co-expressed gene pairs**. A framework for robustly identifying local gene co-expression was developed in this study. The input for this method is a gene expression matrix across individuals (e.g. genes as rows, individuals as columns) including gene coordinates (chromosome and TSS position). In this study,

the input expression matrices were previously pre-filtered, normalised and corrected for covariates (see above). Gene pair Pearson correlation across individuals is then calculated for all gene pairs whose TSS's are <1 Mb apart. This window size is commonly used in genetic studies for assessing cis effects and allows to test the vast majority of genes. For instance, in Geuvadis LCLs 16,840 genes have a neighbour within 1 Mb, out of the 16,907 genes assessed. The total number of gene pairs tested are thus 224,267 gene pairs for the Geuvadis dataset and 350,564 gene pairs for the GTEx dataset across all tissues (e.g. 242,084 for Lung and 192,637 for Muscle Skeletal). Of note, the gene pair correlation is calculated regardless of gene order, transcription directionality and coordinate overlap, thus accounting for all possible co-expression occurring in a certain cis window. To account for the variable number of genes neighbouring in a cis window the following procedure was applied: (i) for each gene, it's cross-individual expression values were shuffled and the correlation with each other gene in the cis window was recalculated 1000 times (while keeping the expression values of the other genes intact); (ii) in each randomisation, the highest correlation value across cis genes is kept in order to build a null distribution; (iii) an empirical p-value ('adjusted p-value') is obtained for each gene pair for the correlation value observed with the real data to be higher than the correlation values obtained in the randomisations. Notably, this procedure adjusts for the total number of neighbouring cis genes while ensuring that the correlation structure between them is unchanged. To control for the total number of genes tested with this approach, the Benjamini–Hochberg procedure was applied on the adjusted p-values corresponding to the highest correlation values observed for each gene. The adjusted p-value corresponding to FDR 1% was kept and any gene pair whose adjusted p-values are below this adjusted p-value was considered significantly co-expressed gene pairs (COPs). Note that in GTEx this procedure is performed for each tissue separately. Extremely high correlation (≥0.99) was found to be mapping artefacts and were removed from the analysis (164 gene pairs in Geuvadis, none for GTEx). Finally, COPs were grouped into a network using igraph v0.7.1 on Python3.6 and plotted with Cytoscape 3.8.0[42].

**Enrichment to functionally and evolutionarily related datasets**. The over-representation of Geuvadis LCL COPs (unfiltered, positive and negative correlation) as functional-related gene pairs was assessed with one-way Fisher's Exact tests to several datasets: (i) genes annotated with the same exact 'Biological Processes' annotation of the GO[20], obtained from Ensembl v98 via the BioMart interface; (ii) genes belonging to the same biological pathway, gathered from KEGG[23] and Reactome[24] through the Ensembl v98 BioMart data mining tool[43] (25 May 2020); (iii) genes belonging to the same human protein complex, gathered from the CORUM 3.0 database[21] and hu.MAP[21,22] (20-April-2020). UniprotKB IDs were converted to Ensembl IDs with the Uniprot ID mapping tool[44]. Moreover, to assess the overrepresentation of COPs in evolutionarily related human genes, a compendium of human paralog gene pairs was gathered from (i) paralog gene pairs from the Ensembl v98 BioMart data mining tool[43], (ii) human gene pairs belonging to the same OrthoMCL v6[45] cluster and (iii) gene pairs belonging to the same group of the Duplicated Genes Database[46] (last update: June 19, 2015). This compendium of 11,304 local (within 1 Mb) paralog gene pairs was used to filter evolutionarily-related genes in downstream analysis.

**COP dataset filtering**. Given that neighbouring paralog genes may display co-expression due to usage of the same or copied regulatory elements, all 11,304 local paralog genes gathered from Ensembl, OrthoMCL and the Duplicated Genes Database (see above) were excluded from analysis (before COP discovery phase). Moreover, only COPs with positive expression correlation were used for molecular feature analysis and subsequent analysis (>90% of COPs from each dataset). This is because positive and negative gene pair correlation may be regulated in distinct manners and the number of negatively correlated genes is too low for an extensive analysis of only negative correlation (e.g. only 7.1% COPs are negatively correlated in Geuvadis LCLs). In addition, only COPs that passed significance cutoffs (FDR 1%) in both gene pair comparisons assessed (i.e. gene1–gene2 and gene2–gene1) were considered (<1% of COPs per dataset were removed in this way). These filters were applied equally to both Geuvadis LCL (from 9384 to 6668 COPs) and all GTEx tissues (from 64,320 to 40,999 distinct COPs across tissues).

**Creation of non-COP and trans-COP controls**. To control for distance effects in local gene pair co-expression, a set of non-COPs was built for each dataset and tissue. For this, for each COP, a gene pair that is not co-expressed (non-COP) but is matched for the distance between TSSs observed in the COP is randomly sampled. The maximum distance discrepancy allowed between a COP and a matching non-COP is 100 bp. Non-COPs are picked randomly without replacement. To ensure that non-COPs are not COPs that simply did not pass FDR 1%, only non-COPs with adjusted p-value > 0.5 were randomly sampled. Given the large number of non-COPs available (~95% of all gene pairs tested), in no case a COP is lost for not having a matching non-COP in Geuvadis and in most GTEx tissues, exceptions being 4 gene pairs in Adipose_Subcutaneous, 8 in Muscle_Skeletal and 6 in Thyroid.

To compare the proportion of eQTL sharing in cis-COPs and trans-COPs in Geuvadis LCLs, a set of trans-COPs was defined, consisting of all gene pairs (regardless of genomic location) that have correlation above 0.143 (the minimum

correlation value for significant cis-COPs) and are not cis-COPs. Then, for each cis-COP, a random trans-COPs with a correlation value at most 5% different from the cis-COP correlation value was selected. 6316 cis-COP/trans-COP matches were liable to be created in this manner.

**Molecular feature metrics and datasets used**. The following metrics were assessed for their potential to regulate local gene co-expression. These were calculated in the same way for the Geuvadis LCL dataset and three GTEx tissues—Muscle Skeletal, Lung and Cells EBV-transformed lymphocytes (equivalent to LCLs)—unless otherwise stated.

(a) *Total CTCF sites*: Total number of CTCF-binding sites found between the TSSs of both genes in the gene pair. Tissue or cell-line-specific data was collected from the ReMap 2018 v1.2 database[47], which includes Encode and other public datasets of ChIP-seq experiments. MYOBLAST data was used as a proxy for the GTEx Muscle Skeletal tissue, whereas Lung and LCL (GM12878) datasets were readily available.

(b) *Inverted CTCF motifs*: Number of CTCF motifs found in opposite orientations (i.e. pairs of '+' and '−' CTCF motifs) between the TSS of both genes in the gene pair. Note that this does directly count the number of CTCF motifs in a convergent orientation, but is rather intended as a proxy for the number of possibilities to form DNA loops. TF motifs mapped to hg19 were downloaded from the MotifMap database[48] in August 2019. The following CTCF motifs were used: 'LM2_CTCF = CTCF', 'M01200 = CTCF', 'M01259 = CTCF' and 'MA0139 = CTCF'.

(c) *Hi-C contacts*: KR normalised Hi-C contacts between the 5 kb bins encompassing the TSSs coordinate of each gene in the gene pair. Data from the GM12878 cell line (LCL) at 5 kb resolution was used[49]. This metric was only calculated for the Geuvadis LCL dataset.

(d) *Enhancer sharing*: Number of common enhancers interacting with both genes in the gene pair. Predicted enhancer–gene interactions were downloaded from the Enhancer Atlas V2.0[50] (March 2020). Data from the GM12878 cell line was used for Geuvadis and GTEx LCLs as well as human 'Skeletal_muscle' and 'Lung' data for the respective GTEx tissues.

(e) *Total enhancers*: Number of distinct enhancers interacting with each gene in the gene pair. If the same enhancer interacts with both genes in the gene pair, it is counted once. Data downloaded from the Enhancer Atlas V2.0, as above.

(f) *Shared TFs*: Number of distinct TF motifs found within ±50 kb flanking the respective TSS of both genes in the gene pair (i.e. a number of distinct TF motifs in common between the two genes). TF motifs mapped to hg19 were downloaded from the MotifMap database[48] in August 2019, and amount to 607 different TF motifs.

(g) *Total TF-binding sites (total TFBS)*: Sum of TF-binding motifs (including redundant and overlapping motifs) within ±50 kb flanking of each gene TSS in the gene pair. All 607 different TF motifs from the MotifMap database were used.

(h) *Expression level difference (Diff. expr. level)*: Relative difference in expression between each gene in the gene pair, calculated as the absolute difference between the average expression level of the genes in the pair, divided by the mean expression level of the pair. For the Geuvadis dataset, mean raw RPKM values were used, whereas for GTEx tissues, median raw TPM values were used.

(i) *Expression coefficient of variation difference (Diff. coef. var.)*: Absolute difference between the expression coefficient of variation of the two genes in the gene pair, divided by the mean coefficient of variation of the gene pair, calculated from cross-individual raw RPKM values. This metric was only calculated for the Geuvadis LCL dataset.

(j) *GO term sharing*: Total number of BP GO terms that are in common between the gene pair. GO ID matching is exact. Gene GO term annotations were downloaded through the Ensembl v98 BioMart data mining tool[43] (December 2019).

(k) *Linkage disequilibrium (LD)*: A proxy for possible LD between the two genes in the gene pair was obtained as the maximum LD between the variants flanking the TSS (±5 kb) of one gene and those flanking the TSS of the other gene in the gene pair. LD was measured as R-squared using vcftools v0.1.15[51]. This metric was only calculated for the Geuvadis LCL dataset.

(l) *eQTL sharing*: Whether the lead cis-eQTL (FDR 5%, Benjamini–Hochberg procedure) of one gene in the pair is nominally significant ($p$-value < 0.05) for association with the other gene in the gene pair. This is tested two-ways and values are attributed as: 0 if no lead eQTL is shared, 1 if the lead eQTL of one gene is shared, 2 if the lead eQTL of each gene is shared in the other gene of the pair. Sharing is only considered if the effect sign (beta sign) is conserved across the pair. Only lead eQTLs of eGenes are considered. eQTL mapping was performed with QTLtools v1.1[41] on the same samples and individuals studied for local gene co-expression. Further details on the eQTL analysis are provided below.

(m) *Co-expressed eGenes*: Whether none, one or both genes in the pair are eGenes, i.e. having a significant cis-eQTL (FDR 5%, Benjamini–Hochberg procedure), using the same cis-eQTLs as in the "eQTL sharing" metric.

For all metrics, missing data and 'NA's were replaced with zeroes. When necessary, annotation coordinates were moved to hg38 (for GTEx tissues) or hg19

(for Geuvadis dataset) using the UCSC liftOver tool[52]. In addition to these metrics, to estimate the distance between gene pairs in terms of centimorgans, genetic distance maps based on recombination estimates from Auton et al. 2007 were used[53]. Using these maps, the absolute centimorgan distance between gene TSSs were interpolated for each Geuvadis LCL COP and non-COP. Moreover, the presence of gene pairs in the same LD block or across LD blocks defined by Berisa and Pickrell 2016[54] was assessed considering gene TSS coordinates.

**Logistic regression models**. Logistic regression for several models of features/metrics liable to explain some of the observed local gene co-expression were performed with custom scripts using R (v3.4.4) project programming language, using the *glm* function. Co-expression was encoded in a binomial format based on co-expression, with COPs regarded as positives (value 1) and distance-matched non-COPs as negatives (value 0, see above for details). The regression models were trained with 80% randomly sampled gene pairs (no replacement), keeping the same sample size between positives and negatives. The ability to correctly predict whether the remainder 20% gene pairs are positives or negatives from the learned models was evaluated for each model independently by measuring the AUC of the ROC curve. The sampling of gene pairs into training or test sets was repeated 50 times and the mean AUC reported, unless otherwise specified.

For simplicity, several related metrics were combined into the same model, and thus the AUCs reported are based on these combinations of metrics: (i) the 'Enhancer' model consists of 'Enhancer sharing' + 'Total enhancers' metrics; (ii) the 'TF' model consists of 'Shared TFs' + 'Total TF-binding sites' metrics; (iii) the 'CTCF' model consists of 'Total CTCF sites' and 'Inverted CTCF motifs'; (iv) the 'Expression' model consists of 'Expression level difference' + 'Expression coefficient of variation difference' metrics (for Geuvadis LCLs only); (v) the 'eQTL sharing' model consists of 'eQTL sharing' and 'Co-expressed eGenes' metrics. All other models are based on a single metric, described above.

**COP sharing across tissues**. Clustering of tissues based on COP sharing was performed in R (v3.4.4) using hclust and Pearson correlation coefficient using a discovery-replication approach: for a pair of tissues (tissue A and tissue B), the percentage of COPs (FDR 1%) of tissue A that are also COPs (FDR 1%) in tissue B (and vice versa) is calculated. This provides a non-symmetrical matrix. Only COPs assessed in both tissue A and B were considered. This was performed before COP dataset filtering.

Local gene pair co-presence is defined as both genes in the pair passing an expression threshold of >0.1 TPM in ≥20% samples and ≥6 reads in ≥20% samples (defined by the GTEx project), which is a prerequisite for COP detection. The calculation of the percentage of tissues where co-present was based on this definition across the 49 GTEx tissues. COP sharing and percentage of tissues where co-expressed was calculated as the number of tissues where the COP is found (FDR 1%) divided by the number of tissues where the pair is co-present. To be able to distinguish simple gene expression tissue-specificity from gene co-expression specificity, only COPs where both genes were present (non-zero expression) in at least three tissues were considered (38,196 out of 40,999). COP sharing categories (conserved, prevalent, specific and unique) were derived from this metric as follows: (i) 'unique COPs', found in only one tissue ($N = 20,781$ across tissues), (ii) 'specific COPs', found in >1 tissue but at most 15% tissues where both genes in the pair are present (range: 2–7 tissues, $N = 10,111$), (iii) 'prevalent COPs', found in >1 tissue and between 15% and 50% tissues ($N = 4863$) and (iv) 'conserved COPs', found in >50% tissues ($N = 2441$).

**eQTL mapping and eQTL-sharing analysis**. Genotypes for the 358 European individuals of the Geuvadis dataset[15] were downloaded from the 1000 Genomes[16] FTP server (http://ftp.1000genomes.ebi.ac.uk/vol1/ftp/release/20130502/). All variant sites with more than two possible alleles or with a MAF below 5% have been removed. In total 6,865,255 variants (6,050,717 SNPs, 809,705 indels, 4833 structural variants) were used in analysis after filters. Genotypes from the GTEx v8 dataset were obtained via dbGaP accession number phs000424.v8.p2, which includes 838 subjects, 85.3% European American, 12.3% African American and 1.4% Asian American. Details regarding donor enrolment, consent process, biospecimen procurement methods and other descriptions is previously described[55]. The phased version of the genotype files was used and only variants with MAF >5% were used for analysis, totalling 6,590,509 variants (6,123,541 SNPs, 466,968 indels).

For each gene expression matrix (normalised and corrected for covariates), i.e. for Geuvadis LCL expression matrix and each GTEx tissue expression matrix, cis-eQTLs were mapped using QTLtools v1.1[41] *cis* function. The mapping window was defined as 1 Mb up-stream and down-stream of the TSS of each gene and in all cases the --normal option was used to enforce gene expression phenotypes to match normal distributions $N(0, 1)$. Two runs of eQTL mapping were produced:

1. identification of eGenes and lead eQTLs: the QTLtools cis --permute 1000 option was used to obtain a lead eQTL for each gene, i.e. the variant with the lowest $p$-value. Of note, the --permute option takes into account the total number of variants tested in each gene by randomly shuffling gene quantifications 1000 times and producing a null distribution of the best association $p$-values obtained in each randomisation. This null distribution is used to fit a beta distribution and thus adjust the nominal $p$-value

obtained with the read data. Furthermore, to account for multiple genes being tested in each gene expression matrix, the FDR was calculated on beta-adjusted nominal $p$-values through the Benjamini–Hochberg procedure and only genes with a lead eQTL with FDR less or equal 0.05 were considered to be eGenes and their lead eQTLs kept.

2. identification of all gene–cis–eQTL associations: the QTLtools *cis* --nominal 0.05 option was used to obtain all gene–cis–variant associations (within the defined cis-window) with nominal $p$-value < 0.05. In this case no adjustment or FDR threshold was applied to these nominal $p$-values, as these were only used to replicate the association of a lead eQTL to neighbouring co-expressed genes (eQTL sharing).

Using both runs of eQTL mapping described above and the sets of COPs (FDR 1%) for each tissue in GTEx and Geuvadis LCL dataset, the lead eQTL (FDR 5%) of each eGene was tested for their ability to associate (i.e. nominal $p$-value < 0.05) with other co-expressed genes (even if these are not eGenes). Lead eQTLs that associate with co-expressed genes in this way are considered as 'shared lead eQTLs'. Importantly, a shared lead eQTL is only considered if the effect sign (beta sign of the regression) is the same for both genes. This definition of eQTL sharing was used for both the molecular feature 'eQTL sharing' and for subsequent functional enrichments of lead eQTLs.

For colocalization analysis, the nominal p-values of all variants 2 Mb up- and down-stream of the TSS of each gene were used (no p-value threshold applied). The COLOC[30] v3.2.1 on R v3.5.1 *abf* function was run for each Geuvadis LCL COP ($N = 6668$) and distance-matched non-COPs ($N = 6668$), i.e. considering each gene in the pair as a trait. Variant MAF calculated with bcftools v1.11 on the 358 Geuvadis samples used for COP identification was used for input. Gene pairs with coloc PP4 (posterior probabilities for hypothesis 4) > 0.5 were considered to share a signal.

**Functional annotations and eQTL enrichments tests**. Several functional annotation datasets were collected from distinct sources:

1. ENCODE[28] functional annotations for LCL (GM12878) downloaded from the EBI FTP server. (i) *Protein binding*: TFBS coordinates from 74 ChIP-seq experiments across distinct TFs; (ii) TSS (promoter region including TSS); (iii) *enhancer* as well as CTCF (CTCF-enriched element) annotations produced by ChromHMM[56] combined with SegWay[57]; (iv) *DNAse*: open chromatin regions predicted through DNAse-seq (narrowPeak, FDR 1%).
2. GENCODE[40] gene body categories of human genes: for Geuvadis LCLs, the coding sequence (CDS), untranslated region (UTR) and intron coordinates for all genes were retrieved from the GENCODE v19 GFF3 file (i.e. the annotation on which gene expression was assessed). For GTEx data, the same gene body regions were retrieved from GENCODE v26 and the 3′UTR and 5′UTR were combined into the same UTR category to reproduce the regions available for the Geuvadis dataset. In all cases, annotations were migrated from hg19 to hg38 when needed using the UCSC liftOver tool[52].
3. Chromatin state annotation from the Roadmap epigenomics Core[29] 15-state model (5 marks) was downloaded in July 2020 for Lymphoblastoid Cells (E116), Lung (E096) and Skeletal Muscle (Male (E107) and Female (E108) samples were concatenated).

The statistical enrichment of the overlap of lead eQTLs—shared lead eQTLs and other lead eQTLs—for each functional annotation was performed in two ways: (i) Direct enrichments: two-way Fisher's exact test between the overlap found for shared lead eQTLs versus the overlap found for other lead eQTLs; (ii) Enrichments over expectation: the assessment of whether eQTLs fall within a certain functional annotations more often than expected by chance was performed with QTLtools v1.3.1[41] *fenrich* function. Briefly, for each eQTL dataset (shared or other) and for each functional annotation, the expected mean number of eQTLs falling within the functional annotations is computed from 10,000 permutations of the annotations found around ±1 Mb of the TSS of each gene. For instance, annotations around gene A are attributed to gene B, accounting for strand orientation and maintaining distances between annotations and TSS identical. In this way, the distribution of functional annotations around genes is unchanged, only their assignment to genes is shuffled through permutations. Then, the observed overlap is compared to the expected one through a two-way Fisher's Exact test. Of note, possible differences between the distribution of functional annotations around eGenes with shared lead eQTLs and those with other lead eQTLs is taken into account in this way, since the expected distributions represent what is expected for the eGenes present in each category. The granularity of the data used consists of eGene–eQTL pairs, i.e. if the same variant is a lead eQTL for two genes, this is counted twice.

For the analysis aggregating pleiotropic lead eQTLs across the 49 GTEx tissues (shared eQTLs and other lead eQTLs) unique eQTLs were used, i.e. if the same eGene–eQTL pair is present in several tissues, this is only counted once. If a lead eQTL is shared in at least one tissue (even if not in other tissues) it is considered as "shared". Overlap of shared eQTLs and other lead eQTLs was compared across functional annotations from the Ensembl v101 Regulatory Build[33] and genic regions from GENCODE v26[40] using BEDTools[58] intersect v2.29.2.

**GWAS summary statistics and trait pleiotropy analysis**. GWAS summary statistics for 35 phenotypes comprising diverse categories such as disease, anthropometric, live-style, cardio-metabolic, blood and neurological traits

(Supplementary Table 2) were downloaded from the UK Biobank project[59] through the Pan UK Biobank resource (Pan-UKB team. https://pan.ukbb. broadinstitute.org. 2020). This resource provides summary statistics of 7221 phenotypes using a generalised mixed model association testing framework and provides results stratified by ancestry groups. For each phenotype, the same set of 28,987,534 variants is used (which includes 96.8% of all distinct GTEx lead eQTLs across tissues), therefore not requiring variant imputation or harmonisation between phenotypes. Only summary statistics for the European ancestry group (EUR, total of 420,531 individuals) were used for each phenotype. Moreover, all phenotypes selected had at least 15,000 cases in the EUR population. Genomic coordinates of genetic variants were lifted over to hg38 using the UCSC liftOver tool. For the pleiotropy analysis, genetic variants were considered to be associated with a phenotype if the (EUR) association $p$-value was below $5e^{-8}$. Quantile–quantile plots of association $p$-values (not filtered) were built against a uniform distribution using the *ggGWAS* R package and the genomic inflation factor was calculated as $\lambda$ (lambda), defined as the median of the resulting chi-squared test statistics divided by the expected median of the chi-squared distribution with one degree of freedom.

**Reporting summary**. Further information on research design is available in the Nature Research Reporting Summary linked to this article.

## Data availability
Data on co-expressed genes and shared eQTLs discovered here are available for consultation and download through the LoCOP database (http://glcoex.unil.ch/) developed here. The list of eQTL–COP–tissue combinations that affect multiple traits generated in this study are provided in Supplementary Data 1. Geuvadis RNA-seq and genotype data are available under accession numbers: EBI ArrayExpress (accession code E-GEUV-1) for RNA-seq data and 1000 Genomes (http://ftp.1000genomes.ebi.ac.uk/vol1/ftp/release/20130502/) for the genotype data. GTEx RNA-seq and genotype data are available from dbGaP (accession: phs000424.v8.p2).

## Code availability
The code used for analysis is available under https://github.com/diogomribeiro/LoCOP[60].

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

## Acknowledgements

O.D. and D.M.R. have been funded by a Swiss National Science Foundation (SNSF) project grant (PP00P3_176977). D.M.R. has also been funded by the European Union's Horizon 2020 research and innovation programme under the Marie Sklodowska-Curie grant agreement No. 885998. The funders had no role in study design, data collection and analysis, decision to publish, or preparation of the manuscript.

## Author contributions

D.M.R. and O.D. developed computational methods. D.M.R. performed experiments and data analysis. S.R. developed the project website. A.R. provided ideas and feedback throughout the project. R.J.H. assisted in genetic variation analysis. D.M.R. and O.D. wrote the manuscript with input from other authors. O.D. and E.T.D. conceived and designed the study.

## Competing interests

E.T.D. is chairman and member of the Board, Hybridstat Ltd. The remaining authors declare no competing interests.
