## [Peer Review File · Nature Communications]

Reviewers' Comments:

Reviewer #1:

Remarks to the Author:

The manuscript "The molecular basis, genetic control, and pleiotropic effects of local gene co-expression" by Ribeiro et al. presents a first comprehensive inspection of co-expression of genes in cis (1MB from each other) throughout the genome in 49 human tissues and LCLs using RNA-seq data from hundreds of individuals (GTEx and Geuvadis for LCLs), and examination of potential shared genetic control. The authors find that local co-expression is prevalent genome-wide (~40% of genes), comparable to previous microarray studies but in a more limited set of tissues. This paper characterizes the molecular features that may be explaining, at least part of the local co-expression using enrichment analysis and logistic regression modeling and AUC. Interestingly they find low regulatory complexity, fewer enhancers and transcription factor binding sites (TFBS), near co-expressed gene pairs (COPs), in particular near COPs found in multiple tissues. They also find that COPs are most highly enriched in the same protein complex and enriched for paralogs. Most notably, they examine for the first time, the role of genetic regulation in controlling local co-expression using eQTLs and suggest that the same genetic variation may be driving 40% of COPs and find that shared eQTLs among co-expressed genes have the strongest predictive power of COPs. Finally, the authors show that eQTLs shared between co-expressed genes tend to have a higher degree of pleiotropic effects on multiple complex diseases and traits, proposing a mechanistic explanation for pleiotropic effects of common variant associations. The findings in this paper further our understanding of the mechanistic underpinnings of local gene co-expression. To increase the impact of this work, the authors have made the COPs publicly available on a web interface (<http://glcoex.unil.ch>).

Overall, the analyses and statistical methods applied in the paper are rigorous, and the authors take caution in correcting for potential confounding effects on COP detection, including hidden covariates of gene expression and genotype principal components that capture differences in ancestral background. I have just a few comments that I think should be addressed to verify that the conclusions are sound, in particular of the shared genetic control among co-expressed genes, and to enhance the biological interpretation of the results.

Major comments:

1. The authors define a 1Mb window around each gene's transcription start site (TSS) for inspecting COPs in cis. Given that a leading hypothesis for co-expression is shared regulation, it would be informative to check whether the significant COPs tend to lie in common linkage disequilibrium blocks, for example as defined in Berisa and Pickrell *Bioinformatics* 2016 (PMID: 26395773), or within the boundaries of adjacent recombination hotspots. The authors show that the extent of genetic linkage between co-expressed gene pairs is similar to that between TSS distance-matched, non co-expressed gene pairs, based on centimorgans. However, this still does not directly address the question of whether COPs tend to lie within the same LD blocks, and to what extent co-expressed gene pairs lie across LD blocks or recombination hotspots.

2. The main comment/criticism I have is in regards to how the authors define shared eQTLs between COPs, which may affect their conclusions about the contribution of shared genetic control to COPs and the enriched functional annotations (e.g., Fig. 3e and Fig. 4). They define shared lead eQTLs as a variant that is the most significant eQTL for one of the genes in the co-expressed gene pair that is an eGene (FDR<5%) and that has a nominal eQTL p-value below 0.05 for the second gene in the pair (results described on page 11). I think the p-value cutoff used for the second gene is too lenient. As shown in the GTEx v6 and v8 papers (PMID: 29022597 and PMID: 33499903), about 93% of SNPs have a nominal eQTL $p < 0.05$. I would recommend defining shared eQTLs either as variants that pass the multiple hypothesis correction for both genes, or at least use a more stringent cutoff for the second gene's eQTL p-value which is closer to the FDR<5% significance level, such as $P < 1E-4$.

The authors further perform colocalization analysis between shared lead eQTLs for the LCL eQTLs from Geuvadis, and find strong statistical support of shared causal mechanisms for about half the shared eQTLs (Suppl Fig. 20). While colocalization methods have been shown to be underpowered, this nevertheless suggests that using an eQTL p-value cutoff of 0.05 for the second gene in the pair may be too lenient.

Furthermore, the authors test whether the shared eQTLs in Geuvadis LCLs and GTEx LCLs are enriched in different functional annotations and find the most significant enrichment in enhancers (Fig. 4a-c). This analysis was done using the more lenient definition of eQTL sharing with the second gene having an eQTL P-value below 0.05. Can the authors check if the results persist if they consider only the shared eQTLs that demonstrate significant colocalization ($P > 0.5$)?

Also, do the authors have an explanation for why they find significant enrichment of shared eQTLs in DNaseI hypersensitive regions and protein binding sites in Geuvadis LCLs but not in GTEx LCLs (Fig. 4a-c)?

3. How does the enrichment of molecular features compare between tissue-specific and tissue-shared COPs?

4. The first time trans-COPs are mentioned in the paper is towards the end of the paper (Page 10, line 254) when comparing the sharing of cis-eQTLs between COPs in cis (± 1 Mb between genes' TSS) to COPs in trans, finding that that eQTL sharing occurs 4-5 times more often for local co-expressed genes than genes that are co-expressed in trans at similar correlation values. I think it would add value to the paper to present trans-COPs earlier in the paper and inspect their properties as well. We would expect trans-COPs to be co-regulated by the same trans-regulator/s, such as a TFs, and would share TFBS in the genes' vicinity. It would also be interesting to test whether they share trans-eQTL variants. Since trans-eQTL discovery is still limited in GTEx, perhaps the authors could test the trans-eQTLs from a blood study of about 31k samples (<https://www.biorxiv.org/content/10.1101/447367v1>).

5. The authors investigate the expression level similarity between pairs of co-expressed genes compared to non-COPs and find that co-expressed gene pairs tend to have similar expression levels and reduced expression variation more so compared to non-COPs (Suppl Fig. 12a). From Suppl Fig. 12b, the genes in the COPs seem to have more lowly expressed genes compared to non-COPs when matching on distance between genes' TSS in the gene pairs. It would be interesting to compare the distribution of gene expression in COPs compared to all genes in the genome not matching on any properties, to check whether co-expressed genes tend to be more lowly expressed than all genes genome-wide.

Also, in the discussion on this finding the authors cite a paper that proposes that selection for reduced expression noise is a cause for grouping of genes along the genome, in particular for essential genes (page 15, line 4012). To follow up on this, it would be interesting for the authors to check if the COPs are enriched for essential genes vs non-COPs.

Minor:

1. Typo on page 24, line 673: 'test' should be 'tested'

2. It would be interesting to know what fraction of the co-expressed gene pairs are adjacent genes versus genes > 1 gene apart.

3. Supplementary Fig. 16, can the authors add to the legend, the definition of some of the less obvious, functional abbreviations, such as 8_ZNF or 11_BivFlnk.

4. The observation that eQTLs that act on multiple genes tend to have a higher degree of pleiotropic effects on multiple complex traits has been recently shown in a paper that analyzed the GTEx v8 data: Barbeira et al., *Genome Biology* 2021 (PMID: 33499903). I would recommend commenting on this in the Discussion.

5. In the discussion, I think the authors should emphasize their finding that co-expressed genes are most strongly enriched in the same protein complexes (OR=12.5) compared to being in the same gene ontology (OR=3.7). This is an interesting observation. The co-regulation might help regulate protein ratio in a complex.

Reviewer #2:

Remarks to the Author:

The paper by Ribeiro et al. investigated the local gene co-expression and their eQTL, epigenetic features across 49 human tissues, estimated the potential molecular mechanisms and relevance on human disease comorbidity and trait pleiotropy. It is a highly comprehensive and interesting study. The authors developed a novel framework to detect local gene co-expression and found that local gene co-expression occurred in 13% to 53% of the genes in the genome. The study offered new insights regarding how gene expression is regulated in genomic regions, and more interestingly new way to explain disease comorbidity and trait pleiotropy.

Several major concerns need to be addressed:

1. Authors collected evolutionarily-related paralog genes and studied their enrichment in three kinds of COPs. It is quite interesting. What is the biological hypothesis and expectation there? Since they used paralog to construct the evolutionarily-related COPs, constructing evolutionarily-unrelated COPs should be easy. Would these evolutionarily-related COPs have eQTL, regulatory regions more conserved than those evolutionarily-unrelated COPs, or not? Would they be more tissue-specific or not? Would they be more involved in trait pleiotropy or not? -- These are questions to think about. Otherwise, the evolutionary aspect seems to be a missed opportunity.

It should be noted that paralog only captured a small part of gene evolution. The best evolution analysis needs to compare genomes and transcriptomes across different species. It might be a big topic for a separate paper.

2. The negative COP is highly surprising, also interesting. Unfortunately, the authors chose to skip them in most of the analyses for the reason of a small number. In fact, 668 in the LCL data is not too small. I don't know what that number is for the other tissues. I don't understand why they cannot be compared with those positive COPs as they did in the functional enrichment analysis. Actually, I am quite curious about how much they differ from positive COP besides the association direction. Their unique genomic features, tissue-specificity, involvement in disease GWAS, trait pleiotropy, etc. are all interesting topics.

It would be nice if they can offer some kind of mechanistic hypothesis as well for the negative correlations.

3. In Figure 3.b, those AUCs might be better to be sorted. The colors of those lines are too similar to tell.

4. They presented data of COP sharing across tissues but did not present tissue-specificity of the eQTL of those COPs, which is equally important.

5. The functional enrichment analysis can be performed on the tissue-specific COP as well.

6. In the Methods, line 449-455, the authors described both PCA and PEER. But they did not mention age. I am confused whether they used both PCA and PEER in adjusting data, whether there is a concern of overcorrection, whether age factor has been properly controlled. Please clarify.

7. 1% is used as the FDR threshold for gene co-expression analysis, is there a consideration of correlation coefficient for a gene pair to be called co-expressed?

8. It is not clear why they used panUK instead of UKbiobank data, which has much larger sample size

and more phenotypes for their trait pleiotropy analysis.

Minor concerns or suggestion for improvement:

1. The authors offered a nice website. The table query function is excellent. It would be better if they can plot the involved genes and SNPs for those COPs with shared SNPs using something like the genome browser's view.
2. The authors listed many different sources of databases for their molecular feature analyses. Can they clarify whether those data align well with EpiMap (<http://compbio.mit.edu/epimap/>), which is commonly used and pretty comprehensive.
3. For cross tissue analysis, the authors found a strong correlation between the numbers of co-expressed genes detected and the RNA-seq sample sizes available per tissue. GTEx brain sample size is relatively small. Associated with that, they detected much fewer COPs in the brain (supplementary figure 4). The eQTL sharing is less affected by the sample size, interestingly. It might be good use try this using larger RNA-seq data sets, such as PsychENCODE or ROSMAP, to maximize the COP and eQTL detection.
4. Since COP is short for "co-expressed gene pair," the authors should use COPs in the paragraphs like around line 133.
5. In Fig.2, the authors listed the number of co-expressed local genes across tissues. It is better to list the total of local genes used for each co-expression analysis after the filtering too.

At the end, I have to acknowledge, this is a long paper with a lot of interesting results. I might overlook or miss some points and make inadequate comments. I would apologize if that happened. Please feel free to offer clarification or counterargument.

REVIEWER COMMENTS

Revision summary:

- First, we would like to mention that we highly appreciate the comments/suggestions from the referees and their interest in our work. As this was a novel research work, many questions remain to be addressed, and we share our enthusiasm with the reviewers for many of the suggestions proposed. We hope to research many of the questions and ideas suggested by the referees in our future work on this topic.
- We hope you will appreciate our effort into incorporating the referee suggestions whenever possible. However, we have already performed many analyses for this study, and it is hard to accommodate all our findings into a cohesive and readable manuscript, as the manuscript is already dense with results and information. Therefore we considered that some suggestions would make the manuscript too complex for readers and decided to add some of them as supplementary notes instead.
- For transparency, we would like to explain that during revision, we spotted two minor issues with the existing Figures (Fig. 3d and Supplementary Fig. 13a) which we have now corrected. These did not impact results, only the figure display.

Reviewer #1 (Remarks to the Author):

The manuscript “The molecular basis, genetic control, and pleiotropic effects of local gene co-expression” by Ribeiro et al. presents a first comprehensive inspection of co-expression of genes in cis (1MB from each other) throughout the genome in 49 human tissues and LCLs using RNA-seq data from hundreds of individuals (GTEx and Geuvadis for LCLs), and examination of potential shared genetic control. The authors find that local co-expression is prevalent genome-wide (~40% of genes), comparable to previous microarray studies but in a more limited set of tissues. This paper characterizes the molecular features that may be explaining, at least part of the local co-expression using enrichment analysis and logistic regression modeling and AUC. Interestingly they find low regulatory complexity, fewer enhancers and transcription factor binding sites (TFBS), near co-expressed gene pairs (COPs), in particular near COPs found in multiple tissues. They also find that COPs are most highly enriched in the same protein complex and enriched for paralogs. Most notably, they examine for the first time, the role of genetic regulation in controlling local co-expression using eQTLs and suggest that the same genetic variation may be driving 40% of COPs and find that shared eQTLs among co-expressed genes have the strongest predictive power of COPs. Finally, the authors show that eQTLs shared between co-expressed genes tend to have a higher degree of pleiotropic effects on multiple complex diseases and traits, proposing a mechanistic explanation for pleiotropic

effects of common variant associations. The findings in this paper further our understanding of the mechanistic underpinnings of local gene co-expression. To increase the impact of this work, the authors have made the COPs publicly available on a web interface (<http://glcoex.unil.ch>).

Overall, the analyses and statistical methods applied in the paper are rigorous, and the authors take caution in correcting for potential confounding effects on COP detection, including hidden covariates of gene expression and genotype principal components that capture differences in ancestral background. I have just a few comments that I think should be addressed to verify that the conclusions are sound, in particular of the shared genetic control among co-expressed genes, and to enhance the biological interpretation of the results.

We thank the referee for the careful reading of the manuscript and the pertinent comments.

Major comments:

1.

The authors define a 1Mb window around each gene's transcription start site (TSS) for inspecting COPs in cis. Given that a leading hypothesis for co-expression is shared regulation, it would be informative to check whether the significant COPs tend to lie in common linkage disequilibrium blocks, for example as defined in Berisa and Pickrell Bioinformatics 2016 (PMID: 26395773), or within the boundaries of adjacent recombination hotspots. The authors show that the extent of genetic linkage between co-expressed gene pairs is similar to that between TSS distance-matched, non co-expressed gene pairs, based on centimorgans. However, this still does not directly address the question of whether COPs tend to lie within the same LD blocks, and to what extent co-expressed gene pairs lie across LD blocks or recombination hotspots.

We thank the referee for the analysis suggestion. To directly address whether COPs tend to lie within the same LD blocks, we used the Berisa & Pickrell 2016 dataset (EUR population) as suggested.

By considering gene TSSs, we found that 87.7% COPs to be in the same LD block, which is as many as distance-matched non-COPs for both Geuvadis LCLs (Supplementary Fig. 14 panel c below). A relatively high number of COPs and non-COPs in the same LD block is expected given that these gene pairs are often found at close distances (main manuscript Fig. 3a). As a comparison, on average 70.1% the gene pairs tested (i.e. within 1Mb distance) are in the same LD block. The same was verified across the 49 GTEx tissues (Response Fig. 1).

We have added a new panel to Supplementary Fig. 14 (see below), as well as corresponding entries in results "Indeed, the genetic linkage of COPs and non-COPs

does not substantially differ, as measured by their TSS centimorgan distance and presence in the same LD block” and methods “Moreover, the presence of gene pairs in the same LD block or across LD blocks defined by Berisa & Pickrell 2016⁵⁴ was assessed considering gene TSS coordinates.”.

Given these results, and together with the results presented in the manuscript: (i) similar centimorgan distance between COPs and non-COPs (Supplementary Fig. 14a,b), (ii) low AUC (0.52) attributed to LD for predicting Geuvadis LCL COPs (main Fig. 3b), even when considering only COPs with distance >200kb (AUC = 0.54, Supplementary Fig. 8), we consider that the consistent comparison of COPs to non-COPs performed across our manuscript reflects a LD-unbiased view of the results.

Supplementary Fig. 14 panel c. [...] number of gene pairs found in the same LD block or across LD blocks (based on Berisa & Pickrell 2016, Methods). For a comparison, the number for all other gene pairs (N = 183748) is shown in yellow.

Response Fig. 1. LD blocks in GTEx COPs. (a) Violin plot including jitter points of the percentage of gene pairs in the same LD block or across LD blocks for all 49 GTEx tissues (i.e. each dot is a tissue), comparing COPs, distance-matched non-COPs and all tested gene pairs (within 1Mb); (b) Number of COPs and distance-matched non-COPs in the same LD block across 49 GTEx tissues.

2.

The main comment/criticism I have is in regards to how the authors define shared eQTLs between COPs, which may affect their conclusions about the contribution of shared genetic control to COPs and the enriched functional annotations (e.g., Fig. 3e and Fig. 4). They define shared lead eQTLs as a variant that is the most significant eQTL for one of the genes in the co-expressed gene pair that is an eGene (FDR<5%) and that has a nominal eQTL p-value below 0.05 for the second gene in the pair (results described on page 11). I think the p-value cutoff used for the second gene is too lenient. As shown in the GTEx v6 and v8 papers (PMID: 29022597 and PMID: 33499903), about 93% of SNPs have a nominal eQTL $p < 0.05$. I would recommend defining shared eQTLs either as variants that pass the multiple hypothesis correction for both genes, or at least use a more stringent cutoff for the second gene's eQTL p-value which is closer to the FDR<5% significance level, such as $P < 1E-4$.

We thank the reviewer for bringing up this remark about the eQTL sharing cutoff, as we agree that this is a point for concern and something we also thought about during our study. First, we would like to put things into context:

We think the value of 93% SNPs having a nominal $p < 0.05$ in the GTEx paper is primarily addressed to the GWAS community. Indeed, interpreting a given GWAS variant using this threshold is risky, as one SNP in GTEx is tested across >40 tissues and on average ~13 genes in the 2Mb cis window (= ~520 tests total), and therefore is almost guaranteed to present at least one $p < 0.05$ with gene expression just because of multiple testing. In contrast, here we only perform a **single test** for eQTL sharing as we know the specific gene-tissue combination that needs testing. In this setting, a threshold of 0.05 is good practice and seems perfectly reasonable to us.

In any case, we felt the need to demonstrate how using more stringent replication p-value cutoff impacts our discovery power. For this we tested two p-value cutoffs, i) $1e^{-4}$, as suggested by the referee, and ii) a 5% FDR threshold (Benjamini-Hochberg procedure). In Geuvadis LCL, a FDR of 5% corresponds to a nominal of 0.025 (i.e. close to the 0.05 used in the manuscript). For the replication cutoff $1e^{-4}$, the number of COPs with eQTL sharing drops from 41.9% to 24.3% (Revision Fig. 2a-b), i.e. we lose 42% of our discoveries. For the replication cutoff 0.025 (5% FDR), the number of COPs with eQTL sharing drops from 41.9% to 38.7% (Revision Fig. 2a,c), i.e. only a slight change.

Revision Fig. 2: eQTL sharing in Geuvadis gene pairs depending on nominal p-value cutoff (a) Cutoff 0.05 (as in the manuscript) (b) Cutoff $1e^{-4}$ (c) Cutoff 0.025 (BH FDR 5% on 13336 tests performed).

To understand how this impacts downstream results, we have redone the molecular feature analysis for Geuvadis LCLs (Revision Fig. 3). Using the $1e^{-4}$ cutoff, we observe a decrease in the AUC of eQTL as a metric predictive of COPs, from 0.74 (main Fig. 3b) to 0.65, meaning that many shared eQTLs we lose using the stringent threshold have predictive power. For the cutoff of 0.025, the AUC remains similar (0.73).

Revision Fig. 3: Geuvadis LCL molecular features depending on nominal p-value cutoff (a) Cutoff $1e^{-4}$; (b) Cutoff 0.025 (BH FDR 5%).

The authors further perform colocalization analysis between shared lead eQTLs for the LCL eQTLs from Geuvadis, and find strong statistical support of shared causal mechanisms for about half the shared eQTLs (Suppl Fig. 20). While colocalization methods have been shown to be underpowered, this nevertheless suggests that using an eQTL p-value cutoff of 0.05 for the second gene in the pair may be too lenient.

Interestingly, for the $1e^{-4}$ cutoff, while the percentage of COPs in eQTL sharing that colocalize increased from 45.6% to 63.1%, the percentage of COPs not in eQTL sharing also increased from 0.9% to 5.6% (Revision Fig. 4a), i.e. with this cutoff, a higher fraction of COPs that are considered not to be in eQTL sharing are actually colocalized, meaning that many well colocalized eQTLs are declared as unshared in this scenario. Conversely, using the 0.025 cutoff did not produce a change compared to using the 0.05 cutoff (i.e. the percentage of COPs not in eQTL sharing kept at 0.9%; Revision Fig. 4b).

Revision Fig. 4: Colocalization of COPs in eQTL sharing. (a) Cutoff $1e^{-4}$; (b) Cutoff 0.025 (FDR 5%).

Furthermore, the authors test whether the shared eQTLs in Geuvadis LCLs and GTEx LCLs are enriched in different functional annotations and find the most significant enrichment in enhancers (Fig. 4a-c). This analysis was done using the more lenient definition of eQTL sharing with the second gene having an eQTL P-value below 0.05. Can the authors check if the results persist if they consider only the shared eQTLs that demonstrate significant colocalization ($PP4 > 0.5$)?

We had previously performed functional enrichments for colocalized shared eQTLs. We found that the functional enrichment of shared eQTLs with $PP4 > 0.5$ is stronger than for shared eQTLs with $PP4 < 0.5$ (Revision Fig. 5), i.e. shared eQTLs with $PP4 > 0.5$ can be seen as a stringent high-confidence dataset of shared eQTLs. **This has been added to Supplementary Fig. 21 as an additional panel**, referenced in the Results section: **“In addition, we find that shared eQTLs with $PP4 > 0.5$ display stronger functional enrichments than those with $PP4 < 0.5$ (Supplementary Fig. 21). Yet, even shared eQTLs with $PP4 < 0.5$ are enriched for various annotations, including enhancers (OR = 2.2, p-value = $1e^{-4}$).”**

However, we decided to use shared eQTLs based solely on the discovery/replication approach (i.e. regardless of *coloc* PP4) for multiple reasons: (i) as exposed by the referee, we consider using *coloc* to be underpowered and in our exploratory study we wanted to keep the highest possible number of ‘viable’ shared eQTLs for downstream analysis; (ii) shared eQTLs with PP4<0.5 also display significant enrichments to enhancers and other features (Supplementary Fig. 21 panel c) and thus may represent *bona fide* variants involved in gene co-regulation; (iii) *coloc* was developed for detecting colocalization between eQTL and GWAS signals and not gene co-expression.

Supplementary Fig. 22 panel c: Geuvadis LCL functional enrichment for shared eQTLs with *coloc* PP4>0.5 (solid color, round points, N = 451) and shared eQTLs with *coloc* PP4≤0.5 (pale color, triangles, N = 2303).

Also, do the authors have an explanation for why they find significant enrichment of shared eQTLs in DNaseI hypersensitive regions and protein binding sites in Geuvadis LCLs but not in GTEx LCLs (Fig. 4a-c)?

Thank you for your remark. To clarify, we would like to point out that GTEx LCLs shared eQTLs (as well as unshared eQTLs) are also significantly enriched for both DNaseI hypersensitive regions and protein binding sites, as seen for Geuvadis LCLs (Revision Fig. 5b). The difference is that for GTEx LCLs we only found a significant difference between shared and unshared eQTLs for the enhancer category (main Fig. 4c, also shown below). We believe this to result from the lower sample size of the GTEx dataset (147 individuals compared to 358 in geuvadis) and the consequent lower eQTL mapping power. Indeed, GTEx LCLs is the 7th GTEx tissue with the lowest percentage of COPs with shared eQTLs (main Fig. 4d). The outcome of this is that only 53 shared eQTLs

overlap with DNase (301 for Geuvadis LCLs), and only 157 overlap with protein binding regions (807 for Geuvadis), which could explain our inability to see a difference between shared and unshared eQTLs for GTEx LCLs.

Revision Fig. 5: Functional enrichment of (a) Geuvadis LCL and (b) GTEx LCL shared and unshared eQTLs. Note that panel a is identical to Figure 4a,b with the addition of the number of eQTLs in each barplot.

3.

How does the enrichment of molecular features compare between tissue-specific and tissue-shared COPs?

In the manuscript we show the molecular features results discriminating for tissue-specific/unique, prevalent and conserved COPs, depending on their presence across the 49 GTEx tissues. These results are shown in Supplementary Fig. 10-12 for LCL, Lung and Muscle skeletal tissues, respectively. These indicate that tissue-shared COPs have stronger effects across all molecular features. We have clarified the text regarding this in the main manuscript **“Indeed, the AUCs for conserved COPs are consistently higher**

than specific COPs for each molecular feature, including a more pronounced depletion of TFBSs and interacting enhancers [...]"

We would also like to mention the response to referee #2, point 5, regarding the functional enrichment analysis of the eQTLs of tissue-specific COPs which may also be relevant in this context.

4.

The first time trans-COPs are mentioned in the paper is towards the end of the paper (Page 10, line 254) when comparing the sharing of cis-eQTLs between COPs in cis (+/- 1Mb between genes' TSS) to COPs in trans, finding that that eQTL sharing occurs 4-5 times more often for local co-expressed genes than genes that are co-expressed in trans at similar correlation values. I think it would add value to the paper to present trans-COPs earlier in the paper and inspect their properties as well. We would expect trans-COPs to be co-regulated by the same trans-regulator/s, such as a TFs, and would share TFBS in the genes' vicinity. It would also be interesting to test whether they share trans-eQTL variants. Since trans-eQTL discovery is still limited in GTEx, perhaps the authors could test the trans-eQTLs from a blood study of about 31k samples (<https://www.biorxiv.org/content/10.1101/447367v1>).

We thank the referee for these remarks. Indeed, we agree that comparing cis-COPs and trans-COPs is of interest and we have performed several analyses to obtain a glimpse of this comparison. However, we consider this a separate narrative than the message we envisaged to pertain in this manuscript. We thus added these new results as Supplementary Note 2 (copied below), referenced in the Results section: **"Supplementary Note 2 compares a subset of the assessed molecular features between cis-COPs and trans-COPs."**

Regarding analysing the sharing of trans-eQTLs using the Vösa *et al.* 2018 eQTLGen dataset (hereby named as "eQTLGen"), we would like to note that eQTLGen only provides data for a subset of 10,319 trans-eQTLs associated to complex traits, which results in an insufficiently number of variants in which to study the propagation of regulatory signals and identify co-regulation of genes genome-wide.

Supplementary Note 2: Molecular feature comparison between cis-COPs and trans-COPs

To investigate how Geuvadis LCL cis-COP molecular features compare to those of trans-COPs, we defined trans-COPs as having correlation above 0.143 (Methods), and trans-non-COPs as gene pairs having low correlation value (i.e. <0.01). Then, for each cis-COP, a random trans-COPs with a similar correlation value (i.e. at most 5% difference) from the cis-COP correlation value was selected. This resulted in 6316 cis-COP/trans-COP matches (and 6316 corresponding cis-non-COPs and trans-non-COPs). Several molecular features such as enhancer sharing and counting CTCF sites between genes can only be performed for cis-COPs. We

thus performed a molecular feature analysis for total TFBS, shared TFs, GO term sharing and difference in expression level/variation.

First we computed the AUCs for discriminating trans-COPs versus trans-non-COPs. We found the expression level difference/variation to have the highest AUC (0.73, Supplementary Note 2 Fig. 1), a value higher than for cis-COPs (0.64). This difference is mostly driven by the fact that trans-non-COPs have more divergent expression than cis-non-COPs (Supplementary Note 2 Fig. 2). In terms of the transcription factor features, these are less discriminative of co-expression for trans-COPs (AUC 0.55) than for cis-COPs (AUC 0.61). However, we still found that trans-COPs display a lower amount of transcription factor presence around the TSS region compared to trans-non-COPs, indicating a similar evolutionary pressure as for local gene co-expression (Supplementary Note 2 Fig. 2).

Next, we directly compared trans-COPs and cis-COPs. Here we find that the main discriminating feature between trans and cis COPs is the TF metrics (AUC = 0.75, Supplementary Note 2 Fig. 1). In fact, while the number of total TFBS is very similar between cis-COPs and trans-COPs, cis-COPs tend to have higher transcription factor sharing (mean 23 for cis-COPs, 16.9 for trans-COPs). However, we observe even higher transcription factor sharing (mean 26.6, Supplementary Note 2 Fig. 2) for cis-non-COPs, which indicates that the distribution of transcription factor binding sites around nearby gene pairs drives the distinction between trans-COPs and cis-COPs. Finally, GO term sharing shows similar levels across categories, indicating that functional similarity pressures are alike for cis-COPs and trans-COPs.

Supplementary Note 2 Fig. 1: Molecular features of 1) trans-COPs versus trans-non-COPs, 2) cis-COPs versus cis-non-COPs and 3) trans-COPs versus cis-COPs. Same metrics and parameters as in the manuscript were used (80% train set, 20% test set). Boxplots are produced from 50 randomisations of the test/training set. N = 6316 for each category.

Supplementary Note 2 Fig. 2: Details of the molecular features across COP and non-COP datasets. **a** expression level difference; **b** expression coefficient of variation difference; **c** total transcription factor binding sites (around 50Kb of TSS); **d** shared TFs, i.e. number of distinct transcription factor motifs shared between the gene pair.

5.

The authors investigate the expression level similarity between pairs of co-expressed genes compared to non-COPs and find that co-expressed gene pairs tend to have similar expression levels and reduced expression variation more so compared to non-COPs (Suppl Fig. 12a). From Suppl Fig. 12b, the genes in the COPs seem to have more lowly expressed genes compared to non-COPs when matching on distance between genes' TSS in the gene pairs. It would be interesting to compare the distribution of gene expression in COPs compared to all genes in the genome not matching on any properties, to check whether co-expressed genes tend to be more lowly expressed than all genes genome-wide.

As suggested by the referee, **we have added the expression distribution for all tested gene pairs to Supplementary Fig. 13 (before revision, Supplementary Fig. 12) panel b** (copied below). This shows that COPs' mean expression (1.94 log₂(RPKM), N = 6668) is lower than other gene pairs tested (2.13 log₂(RPKM), N = 183748). We would like to point out that this is a relatively small difference in mean expression levels which may not be meaningful. Moreover, we show in Supplementary Fig. 13 panels b and c that controlling for expression level does not change the expression-related molecular feature results, which was the point of this control analysis.

Supplementary Fig. 13 update. [...] (b) details of the mean expression level control (between the two genes in the pair) used for this specific analysis. After controlling for both distance (at most 5% difference in distance allowed between COP and matching non-COP) and mean expression level (at most 10% difference allowed), the mean expression distribution between COPs and non-COPs clearly matches (right plot). In the step of picking non-COPs matched for both distance and expression level, 545 COPs were lost. **For a comparison, the mean expression distribution for all other gene pairs (N = 183748) is shown in blue;** [...]

Also, in the discussion on this finding the authors cite a paper that proposes that selection for reduced expression noise is a cause for grouping of genes along the genome, in particular for essential genes (page 15, line 4012). To follow up on this, it would be interesting for the authors to check if the COPs are enriched for essential genes vs non-COPs.

We thank the referee for the interesting suggestion. To assess if Geuvadis LCL COPs have higher gene essentiality than non-COPs, we first used the LOEUF (loss-of-function observed/expected upper bound fraction) metric from the Gnomad database (<https://gnomad.broadinstitute.org/>, described in PMID: 32461654). A gene was considered 'essential' if its LOEUF value was in the first decile, i.e. the subset of the most essential genes (N = 1920). While we only found 64 COPs composed of essential genes, this compares to 34 non-COPs, indicating that COPs may have a slightly higher proportion of essential genes than non-COPs (Revision Fig. 6, Fisher's Exact test p-value = 0.003, OR = 1.9).

To try to confirm this trend, we used a list of 678 human essential genes based on CRISPR/SpCas9 Knockout Screens from Hart *et al.* 2017 (PMID: 28655737). We found 29 COPs composed of essential genes compared to 17 non-COPs, yet the sample size is insufficient to be confident of this result (Fisher's Exact test p-value = 0.1, OR = 1.7).

Overall, while these results may indicate COPs could be enriched for essential genes when compared to non-COPs, the actual number of COPs that are composed of essential genes seems too small for us to make a bold and confident claim. To avoid over-interpreting this weak signal, we decided not to include this analysis in the manuscript.

Revision Fig. 6. Number of COPs and non-COPs composed of essential genes based on the LOEUF metric. N = 3343 for each category, after excluding genes with no LOEUF value available.

Minor:

1. Typo on page 24, line 673: 'test' should be 'tested'

Thank you for noticing. The text now reads **“Furthermore, to account for multiple genes being tested [...]**”

2. It would be interesting to know what fraction of the co-expressed gene pairs are adjacent genes versus genes >1 gene apart.

We thank the reviewer for this interesting question. Similar to what was observed for the distance distribution (Fig. 2a), COPs are more often formed between the nearest gene neighbours. **We added this information as additional panels on Supplementary Fig. 1 and reference it in the Results section: “A higher proportion of protein coding genes (54.8%) is co-expressed compared to lincRNAs (43.5%), and as much as 53% COPs are formed between the nearest neighbours (Supplementary Fig. 1).”**

Supplementary Fig. 1 update. [...] (c) distribution of the number of genes (TSSs of tested genes) found between gene pairs, considering all genes regardless of strand. 41% of Geuvadis COPs are formed between the nearest neighbours; (d) considering only positively stranded genes, as an example of considering gene neighbours only if being on the same strand. 53% are formed between the nearest neighbours.

3. Supplementary Fig. 16, can the authors add to the legend, the definition of some of the less obvious, functional abbreviations, such as 8_ZNF or 11_BivFlnk.

We added the following to the legend of Supplementary Figures 17 to 20 (before revision, Supplementary Figures 16 to 19): **“Annotation legend: 1_TssA: Active TSS, 2_TssAFlnk: Flanking Active TSS, 3_TxFlnk: Transcr. at gene 5' and 3', 4_Tx: Strong transcription, 5_TxWk: Weak transcription, 6_EnhG: Genic enhancers, 7_Enh: Enhancers, 8_ZNF/Rpts: ZNF genes & repeats, 9_Het: Heterochromatin, 10_TssBiv: Bivalent/Poised TSS, 11_BivFlnk: Flanking Bivalent TSS/Enh, 12_EnhBiv: Bivalent Enhancer, 13_ReprPC: Repressed PolyComb, 14_ReprPCWk: Weak Repressed PolyComb, 15_Quies: Quiescent/Low.”**

4. The observation that eQTLs that act on multiple genes tend to have a higher degree of pleiotropic effects on multiple complex traits has been recently shown in a paper that analyzed the GTEx v8 data: Barbeira et al., Genome Biology 2021 (PMID: 33499903). I would recommend commenting on this in the Discussion.

Thank you for this remark. Barbeira *et al.* 2021 was published just as we submitted our paper.

We modified the discussion to incorporate this study: **“Recent studies also found that eQTLs that affect the expression of multiple genes display high complex trait pleiotropy^{25,39}. Our findings provide further insights into the molecular underpinnings of pleiotropy [...]**”

5. In the discussion, I think the authors should emphasize their finding that co-expressed genes are most strongly enriched in the same protein complexes (OR=12.5) compared to being in the same gene ontology (OR=3.7). This is an interesting observation. The co-regulation might help regulate protein ratio in a complex.

Thank you for the suggestion. Indeed we also found the high enrichment of COPs in the same protein complex striking (when compared to same pathway or same gene ontology term).

We have added the following remark discussion: **“Interestingly, we found that COP genes are enriched in belonging to the same biological pathway (OR = 2.6, Supplementary Fig. 3), biological process GO term annotation (OR = 3.9) and protein complex (OR = 12.5). The particularly high enrichment in COPs encoding proteins belonging to the same protein complex could indicate that gene co-regulation may aid the maintenance of stoichiometry in protein complexes. Our finding that COPs have more similar expression levels and variation compared to non-COPs corroborates this notion.”**

Reviewer #2 (Remarks to the Author):

The paper by Ribeiro et al. investigated the local gene co-expression and their eQTL, epigenetic features across 49 human tissues, estimated the potential molecular mechanisms and relevance on human disease comorbidity and trait pleiotropy. It is a highly comprehensive and interesting study. The authors developed a novel framework to detect local gene co-expression and found that local gene co-expression occurred in 13% to 53% of the genes in the genome. The study offered new insights regarding how gene expression is regulated in genomic regions, and more interestingly new way to explain disease comorbidity and trait pleiotropy.

We thank the referee for the interest in our work and for taking the time to carefully read and review our manuscript.

Several major concerns need to be addressed:

1.

Authors collected evolutionarily-related paralog genes and studied their enrichment in three kinds of COPs. It is quite interesting. What is the biological hypothesis and expectation there? Since they used paralog to construct the evolutionarily-related COPs, constructing evolutionarily-unrelated COPs should be easy. Would these evolutionarily-related COPs have eQTL, regulatory regions more conserved than those evolutionarily-unrelated COPs, or not? Would they be more tissue-specific or not? Would they be more involved in trait pleiotropy or not? -- These are questions to think about. Otherwise, the evolutionary aspect seems to be a missed opportunity.

It should be noted that paralog only captured a small part of gene evolution. The best evolution analysis needs to compare genomes and transcriptomes across different species. It might be a big topic for a separate paper.

We thank the referee for the interest in this matter. First, we want to be clear that we only included paralog genes in the COP discovery (the first results section). For the reasons exposed below, in most analysis in the paper we have in fact used evolutionarily-unrelated COPs, i.e. we excluded paralog gene pairs when measuring the molecular features, mapping shared eQTLs and performing pleiotropy analysis.

The decision to exclude paralogs in downstream analysis came after feedback from colleagues (experts in gene evolution), which pointed out that the molecular features we were studying could be heavily affected by neighbouring paralog gene pairs containing the same regulatory elements (e.g. tandem gene duplication including promoter and enhancer regions). In other words, paralog gene pairs are expected to have a particular gene expression regulation, and since we are interested in studying gene co-expression regulation genome-wide, paralog gene pairs should be excluded from analysis. Given this, we are reluctant to include any results for paralogs regarding molecular features and eQTL sharing in the manuscript, as an adequate study of these would require specialised analysis (e.g. creating sets of non-COPs composed of paralog genes, potential comparisons across species). While the study of paralog gene evolution co-expression is an interesting research topic, this is out of scope of the current manuscript.

We have clarified the text regarding paralog removal in the Results section:

“Importantly, since paralog genes may display co-expression due to particular circumstances, such as the use of the same or a duplicated regulatory element, **all gene pairs composed of paralog genes** were excluded from this and **all** subsequent analyses **to exclude this potential bias** (Methods).”

2.

The negative COP is highly surprising, also interesting. Unfortunately, the authors chose to skip them in most of the analyses for the reason of a small number. In fact, 668 in the LCL data is not too small. I don't know what that number is for the other tissues. I don't understand why they cannot be compared with those positive COPs as they did in the functional enrichment analysis. Actually, I am quite curious about how much they differ

from positive COP besides the association direction. Their unique genomic features, tissue-specificity, involvement in disease GWAS, trait pleiotropy, etc. are all interesting topics.

It would be nice if they can offer some kind of mechanistic hypothesis as well for the negative correlations.

We agree with the referee that negatively correlated COPs are interesting to study. We have thus performed several analyses for negatively correlated COPs which we added to the manuscript as Supplementary Note 1 (pasted below), referenced in the Results: **“Moreover, for simplicity, in all analyses we focused on COPs with positive expression correlation, for which we have approximately an order of magnitude more COPs than for negative correlation. Results on negatively correlated COPs are summarized on Supplementary Note 1.”**

Supplementary Note 1: Analysis of negatively-correlated COPs

To compare the molecular feature signature between positively and negatively correlated COPs, we split these two categories of COPs and created a distance-matched set of non-COPs for each category. Overall, we found a similar molecular feature signature between positive and negative COPs (Supplementary Note 1 Fig. 1 and 2). Main differences are *(i)* the AUC for ‘Expression’ (i.e. expression level difference and expression coefficient of variation difference) is lower for negative COPs, as expected for negative correlation; *(ii)* the AUC of LD is higher for negative COPs, indicating negative COPs are more genetically linked than expected; *(iii)* the AUC for eQTL sharing (regardless of effect sign) is high for negative COPs, however this could partially driven by an higher LD in negative COPs. Notably, the decrease of the regulatory complexity compared to non-COPs is still observed for negative COPs, in particular, a lower number of enhancers (Supplementary Note 1 Fig. 2).

Supplementary Note 1 Fig. 1. Geuvadis LCL boxplots of the AUC values obtained for each molecular feature separated by positively and negatively correlated COPs. Values below the boxplot represent the mean over the 50 randomisations. Note that positive correlation and negative correlation datasets were matched for distance distribution separately (i.e. non-COPs match appropriately each dataset of COPs). Note: for positive correlation eQTL sharing is only considered if the effect sign is matched, but for negative correlation consistency of effect sign was not required.

Supplementary Note 1 Fig. 2. Molecular feature boxplots comparing positively and negatively correlated COPs for (a) total enhancers, (b) linkage disequilibrium (LD) measured as R^2 , (c) expression level difference and (d) eQTL sharing. Values next to the boxplots represent the mean. P-values were obtained from two-tailed Wilcoxon signed-rank tests. Note: for positive correlation eQTL sharing is only considered if the effect sign is matched, but for negative correlation consistency of effect sign was not required.

Next, we analysed negatively correlated COPs in GTEx, where we find a total of 8,527 distinct negative COPs. This compares to 64,320 distinct positive COPs. Regarding tissue-specificity, positively correlated COPs show to be more widespread across tissues than negatively correlated COPs (Supplementary Note 1 Fig. 3). On average, positively correlated COPs are present in 3.2 tissues, whereas negative COPs are present in only 1.5 tissues. This indicates that negative COPs may be less relevant across human tissues.

Supplementary Note 1 Fig. 3. COP tissue conservation comparing negatively and positively correlated COPs. (a) distribution of the number of tissues where COP is present; (b) percentage of tissues where COP is found, out of all tissues where the gene pair was assessed. In this panel only gene pairs present in >5 tissues were considered, in order to exclude cases where present is 100% while in fact the gene pair was only assessed in a few tissues.

Finally, following up on the finding that negatively correlated COPs also display high eQTL sharing (albeit with opposite sign effect), we compared the functional enrichment of shared lead eQTLs between positively and negatively correlated COPs (Supplementary Note 1 Fig. 4). The main difference between positive and negative eQTLs is a higher enrichment of negative eQTLs in the UTR region of genes (Supplementary Note 1 Fig. 4). Otherwise, negative-related eQTLs display a similar enrichment against the expected background as positive-related eQTLs, such as a high enrichment in enhancer, protein binding and DNase regions.

Supplementary Note 1 Fig. 4. Comparison of functional enrichment of shared eQTLs in positively and negatively correlated COPs. (a) overlap enrichment of Geuvadis LCL lead shared eQTLs associated with positive COPs (solid color, round points) and lead shared eQTLs associated with negative COPs (pale color, triangles) in Encode LCL functional annotations and Gencode gene body categories. Odds ratios are calculated based on the observed versus expected overlap between eQTLs and each functional annotation. Error bars are from 10000 QTLtools fenrich permutations. The right part of the plot denotes the percentage of overlap between eQTLs and each functional annotation; (b) Fisher's exact test odds ratio and p-value for the enrichment of positive-related eQTLs in each functional annotation, compared to negative-related eQTLs. Error bars are 95% confidence intervals.

Overall, we find negative COPs to be much less numerous than positive COPs, and their lower tissue-conservation suggests a less important role for this category of COPs. Yet, a similar molecular feature signature and the finding that shared eQTL in negative COPs fall in regulatory regions suggests this negative correlation, like positive correlation, may still be regulated by regulatory elements and genetic variation. Mechanisms such as enhancers or TFBS with repressing and activating activity for different genes could play a role in the regulation of negatively correlated COPs.

3.

In Figure 3.b, those AUCs might be better to be sorted. The colors of those lines are too similar to tell.

Thank you for the feedback. **We have improved Fig. 3b as suggested by the referee, as well as increased the size of the lines to improve visibility.**

Fig. 3 panel b update.

4.

They presented data of COP sharing across tissues but did not present tissue-specificity of the eQTL of those COPs, which is equally important.

Thank you for your remark. **We have a new Supplementary Fig. 23** describing the tissue-specificity of eQTLs, referenced on the Results section: **“On average, a COP in eQTL sharing is associated with the same lead eQTL in 21.8% of the tissues where present (considering COPs in >5 tissues, Supplementary Fig. 23).”**

Supplementary Fig. 23. Replication of shared lead eQTL-COPs across tissues.

Percentage of tissues where COP associates with the same shared lead eQTL, out of all tissues where the COP is present. Only COPs present in >5 tissues were considered (N = 4298), in order to exclude cases where for example the eQTL is associated in 100% tissues while in fact the COP only occurs in a few tissues. On average, a COP is associated with the same lead eQTL in 21.8% of the tissues. When several eQTLs are available for a COP, we consider only the most shared eQTL.

5.

The functional enrichment analysis can be performed on the tissue-specific COP as well.

As suggested by the referee, we performed functional enrichment of lead-eQTLs from COPs that are LCL-specific to lead-eQTLs from tissue-shared COPs. However, we would like to point out that functional annotations such as enhancers, protein binding and DNase sites are widely different between tissues (e.g. an enhancer active in a tissue may not be active in another), thus the expectations from this analysis for tissue-conserved COPs are not clear to us.

We found that eQTLs associated with LCL-specific and tissue-conserved COPs display similar enrichment against the background (Revision Fig. 7a), including enrichment in enhancer, dnase and protein binding sites, as seen previously. Directly comparing LCL-specific and tissue-conserved categories shows that LCL-specific lead-eQTL-COPs are only slightly more enriched in enhancers and protein binding sites than tissue-conserved (Fisher's exact test OR: 1.3 - 1.5, p = 0.05, Revision Fig. 7b). Given the lack of evident differences between the two categories, we decided not to include this analysis in the manuscript.

Revision Fig. 7. Functional enrichment of lead-eQTLs in Geuvadis LCLs specific and conserved COPs. (a) enrichment against background; top: LCL-specific (N = 789), bottom: tissue-conserved (i.e. >50% tissues, including LCL, N = 574); (b) direct enrichment between LCL-specific and tissue-conserved COPs. Note that eQTL sharing was not required to be included in the categories (the results when including only shared lead-eQTLs are nearly identical, data not shown). Functional annotations are specific to LCLs.

6.

In the Methods, line 449-455, the authors described both PCA and PEER. But they did not mention age. I am confused whether they used both PCA and PEER in adjusting data, whether there is a concern of overcorrection, whether age factor has been properly controlled. Please clarify.

As stated in the methods, we accounted for sex, subpopulation structure (ancestry), and unknown technical/experimental variables. For the latter, we used PCA for the Geuvadis dataset and PEER factors for GTEx dataset. Both methods are commonly used in eQTL analysis and have been shown to produce similar results (e.g. Cuomo et al. 2021 <https://doi.org/10.1101/2021.01.20.427401>). The reason for using PEER for the GTEx dataset is that we decided to use the exact same covariates as used in the GTEx v8 paper (PMID: 32913098, dbGaP Study Accession: phs000424.v8.p2), i.e. to the best of our knowledge, the best set covariates to be used for this dataset. These include a variable number of PEER factors for each tissue (depending on sample size), sequencing platform, PCR usage, and the sex of the samples. While these covariates did not explicitly account for age, we consider that the covariates accounted for in our

study, in particular the PEER and PCA principal components, should properly account for most confounding variables.

To improve clarity in the manuscript regarding the methods used, we clarified the usage of PCA for Geuvadis and PEER for GTEx in the results section: “In addition, we extensively accounted for known (e.g. sex, subpopulation structure) and unknown confounding factors (using **PCA for the Geuvadis dataset** and **PEER¹⁴ for the GTEx dataset**; Methods).”

7.

1% is used as the FDR threshold for gene co-expression analysis, is there a consideration of correlation coefficient for a gene pair to be called co-expressed?

To clarify, we did not apply any correlation coefficient cutoff in order to determine COPs. Our approach consists of shuffling the expression values of a gene across the individuals 1000 times and measure correlation (while keeping the expression of all other genes in the cis window intact). This means that each gene has it's own null distribution of correlation values (which varies with the number of cis genes and their inner degree of correlation) to determine significance for being a COP. Thus, the correlation level that is high 'enough' to derive a COP varies per gene (as well as for the sample size of the dataset). For instance, in the Geuvadis dataset, a correlation value of at least +/-0.14 is required to approach significance level.

One of the reasons to choose a permutation-based approach for determining COPs was to avoid defining a fixed correlation value threshold, as done in previous studies (e.g. Soler-Oliva *et al.* 2017 PMID: 28902867 used a minimum correlation value of 0.15). However, as we understand that defining a minimum correlation value can be desirable for some analysis, **we added a 'minimumCorrelation' option in our method** so that the user can define a minimum correlation level for defining COPs.

8.

It is not clear why they used panUK instead of UKbiobank data, which has much larger sample size and more phenotypes for their trait pleiotropy analysis.

Thank you for pointing this out. We would like to clarify that the Pan UKBB resource (<https://pan.ukbb.broadinstitute.org/>) is derived from UK biobank data. Simply, the Pan UKBB team performed genetic analysis per population, thus providing association p-values for each population. As most Geuvadis and GTEx individuals have European ancestry, and the most UK biobank (and thus Pan UKBB) individuals also have European ancestry (total of 420,531 individuals at the time of the analysis performed), the choice of Pan UKBB European data is adequate. As far as we are aware, the Pan UKBB GWAS results do not entail a lower sample size than UK biobank data (e.g. from that of Neale's lab <http://www.nealelab.is/uk-biobank>), other than the decreased sample size due to splitting populations with different ancestries.

Regarding the choice of phenotypes, we wanted to select only a few dozen complex traits (with >15.000 cases in EUR population) and thus the Pan UKBB (containing 7,221 phenotypes) was deemed more than sufficient for our purposes. In future works involving more traits we will re-evaluate which resource is better suited to our needs.

To clarify that the Pan UKBB is composed of UK biobank data, we added the following to the Results section: “For this, we first collected GWAS summary statistics for 35 traits from the PanUK Biobank (**Pan-ancestry genetic analysis of the UK Biobank**). [...]”

Minor concerns or suggestion for improvement:

1. The authors offered a nice website. The table query function is excellent. It would be better if they can plot the involved genes and SNPs for those COPs with shared SNPs using something like the genome browser’s view.

We appreciate the suggestion made by the referee. We agree that a genome browser view would be a great addition to the website. However, after careful consideration, we realised that implementing a dynamic genome browser containing tens of thousands of associations between genes as well as genetic variants is a very challenging task. For instance, even major resources such as the GWAS catalog (<https://www.ebi.ac.uk/gwas/>) or GTEx portal do not provide such functionality.

Considering the effort already made to implement a functional database of local gene co-expression and the lack of consensus visualization for 2D data (many genes vs many SNPs), we decided not to undertake such an additional large task at this moment but will instead keep the suggestion for future versions of the database.

2. The authors listed many different sources of databases for their molecular feature analyses. Can they clarify whether those data align well with EpiMap (<http://compbio.mit.edu/epimap/>), which is commonly used and pretty comprehensive.

We thank the referee for pointing us towards the EpiMap resource. The paper presenting the EpiMap resource (PMID: 33536621) was published only after submission of our manuscript. To verify if our results stand also on the EpiMap dataset, we performed the functional enrichment analysis for Geuvadis LCL (GM12878). The results with the EpiMap annotations (Revision Fig. 8) are very similar to those of the RoadMap epigenomics (Supplementary Fig. 17). Namely, we found a strong enrichment for shared eQTLs to fall on enhancer regions. Overall, **our results replicate very well on the EpiMap dataset**, as expected, since the EpiMap data includes the Roadmap Epigenomics dataset and as well as data from other sources.

Revision Fig. 8. Functional enrichments of Geuvadis LCL shared eQTLs using EpiMap annotations. **a** overlap enrichment of shared lead eQTLs (solid color, round points) and other lead eQTLs (pale color, triangles). Odds ratios are calculated based on the observed versus expected overlap between eQTLs and each functional annotation. Error bars are from 10000 fenrich permutations. The right part of the plot denotes the percentage of overlap between eQTLs and each functional annotation; **b** Fisher's exact test odds ratio and p-value for the enrichment of shared lead eQTLs in each functional annotation, compared to other lead eQTLs. Error bars of odds ratio are 95% confidence intervals.

3. For cross tissue analysis, the authors found a strong correlation between the numbers of co-expressed genes detected and the RNA-seq sample sizes available per tissue. GTEx brain sample size is relatively small. Associated with that, they detected much fewer COPs in the brain (supplementary figure 4). The eQTL sharing is less affected by the sample size, interestingly. It might be good use try this using larger RNA-seq data sets, such as PsychENCODE or ROSMAP, to maximize the COP and eQTL detection.

We thank the referee for the dataset suggestions. As the referee pointed out, due to their large sample in the brain these datasets would be complementary to GTEx where brain regions have small sample sizes. We will take this into account in follow-up studies.

4. Since COP is short for "co-expressed gene pair," the authors should use COPs in the paragraphs like around line 133.

We thank the referee for noticing this. We have modified in results and methods section several instances, replacing “co-expressed gene pairs” with “COPs”, including in line 133: “We identified 64,320 distinct **COPs** across tissues (FDR 1%).”

5. In Fig.2, the authors listed the number of co-expressed local genes across tissues. It is better to list the total of local genes used for each co-expression analysis after the filtering too.

We agree and thank the referee for the suggestion. **We have added a new supplementary figure to the manuscript** with the number of COPs per tissue after filtering, referenced in the Results section: “**After these filters, 6668 Geuvadis COPs and 40999 distinct GTEx COPs (number of COPs per tissue on Supplementary Fig. 6) were kept for analysis.**”

Supplementary Figure 6. Number of COPs identified in each GTEx tissue after paralogs and positive correlation filters.

At the end, I have to acknowledge, this is a long paper with a lot of interesting results. I might overlook or miss some points and make inadequate comments. I would apologize if that happened. Please feel free to offer clarification or counterargument.

Thank you for your comment. We agree that this is a long paper, moreover, many interesting questions stem from this study, such as the ones suggested by the referees. We have thus had to make difficult choices on what to include and exclude in the manuscript, in a way that the manuscript would still be readable and with a single storyline.

Reviewers' Comments:

Reviewer #1:

Remarks to the Author:

The authors have done careful work to sufficiently address all of my comments. I have just two very minor comments:

1. I would add the word 'tissue-specific' in front of 'specific' in the following two sentences on page 9: "we observed a clear trend in which conserved COPs generally have higher AUCs across molecular features than specific or unique COPs (Supplementary Fig. 10). Indeed, the AUCs for conserved COPs are consistently higher than specific COPs for each molecular feature,..."

2. I think Supplementary Fig. 1 will be easier to read as a histogram vs. the current density plot, since the x-axis contains discrete numbers.

Reviewer #2:

Remarks to the Author:

I'd like to thank the authors for spending time to generate many new results and figures. Most of my concerns have been properly addressed. Two minor issues:

1. For my minor point 6, why not consistently use PEER for the two data. Authors only explained why they use PEER on GTEx. Even though one paper showed that they produce similar results, it may not be the case for all data. PCA was shown to perform less well for hidden variables, I recall.

Moreover, though age is expected to be correlated with one of PEER/PCA factors, it is better to do a correlation test to confirm.

2. For my previous minor point 7, correlation coefficient issue. I understand the advantages of using permutation, but it is still better to provide a smallest correlation coefficient for that called signals, for both GTEx and Geuvadis data.

3. Regarding the negative COPs, the authors concluded "Overall, we find negative COPs to be much less numerous than positive COPs, and their lower tissue-conservation suggests a less important role for this category of COPs." I think, this is wrong. In contrast, it may indicate negative COPs are more tissue-specific, as their results clearly showed. Tissue-specific regulation is no less important than universal regulation.

The sentence "This indicates that negative COPs may be less relevant across human tissues." has the same problem.

REVIEWER COMMENTS

Reviewer #1 (Remarks to the Author):

The authors have done careful work to sufficiently address all of my comments. I have just two very minor comments:

1. I would add the word 'tissue-specific' in front of 'specific' in the following two sentences on page 9: "we observed a clear trend in which conserved COPs generally have higher AUCs across molecular features than specific or unique COPs (Supplementary Fig. 10). Indeed, the AUCs for conserved COPs are consistently higher than specific COPs for each molecular feature,..."

We thank the referee for the suggestion. The text now reads "we observed a clear trend in which conserved COPs generally have higher AUCs across molecular features than **tissue-specific** or unique COPs (Supplementary Fig. 10). Indeed, the AUCs for conserved COPs are consistently higher than **tissue-specific** COPs for each molecular feature"

2. I think Supplementary Fig. 1 will be easier to read as a histogram vs. the current density plot, since the x-axis contains discrete numbers.

We thank the referee for the suggestion. **We have updated Supplementary Fig. 1c,d accordingly:**

Reviewer #2 (Remarks to the Author):

I'd like to thank the authors for spending time to generate many new results and figures.

Most of my concerns have been properly addressed. Two minor issues:

1. For my minor point 6, why not consistently use PEER for the two data. Authors only explained why they use PEER on GTEx. Even though one paper showed that they produce similar results, it may not be the case for all data. PCA was shown to perform less well for hidden variables, I recall.

Moreover, though age is expected to be correlated with one of PEER/PCA factors, it is better to do a correlation test to confirm.

We understand the reviewer's concern and therefore had a close look at this. To compare our Geuvadis results with PEER covariate correction, we performed COP discovery on a Geuvadis PEER-corrected gene expression matrix (50 PEER factors, to compare with the 50 PCA PCs used in the manuscript). The figure below shows the gene pair correlation values across all 224,267 gene pairs assessed. We clearly observe consistency in correlation values between the PCA and PEER approaches, particularly for COPs (Pearson correlation coefficient = 0.92, P-value < $2.2e^{-16}$). Moreover, 87% of the Geuvadis COPs defined in the manuscript display a permutation p-value < 0.05 with the PEER-corrected matrix (note that this is a stochastic process). This indicates that using either PCA or PEER correction produces very similar results in this dataset.

Revision Fig. 1: Comparison of gene pair correlation between a PCA-corrected gene expression matrix (50 PCA PCs) and PEER-corrected matrix (50 PEER factors). COPs are defined as in the manuscript for the PCA-corrected matrix.

Regarding using age as covariate, age information is not available for Geuvadis individuals (which are part of the 1000 Genomes project), as such data was explicitly not gathered for 1000G samples. More information can be found here:

<https://www.internationalgenome.org/faq/can-i-get-phenotype-gender-and-family-relationship-information-samples/>

For GTEx, an analysis of the correlation of the PEER factors with multiple covariates (including age) was performed in the last main GTEx v8 manuscript (PMID: 32913098, Fig S4, copied below). In this analysis they show that age shows some correlation with PEER factors across tissues. We reiterate that when performing eQTL mapping in GTEx v8, the GTEx consortium did not explicitly include age as a covariate.

Fig. S4. PEER factors for eQTL mapping. Proportion of expression variance captured by the PEER factors computed for each tissue (R^2 , top bar), and proportion of variance (adjusted R^2) removed by the PEER factors explained by known sample and donor covariates. Each cell shows the total proportion of variance removed by all PEER factors. Only covariates with $\geq 0.05 R^2_{adj.}$ in any tissue are shown. Tissues and covariates are ordered based on hierarchical clustering with average Euclidean distance. Gray cells indicate unavailable data.

2. For my previous minor point 7, correlation coefficient issue. I understand the advantages of using permutation, but it is still better to provide a smallest correlation coefficient for that called signals, for both GTEx and Geuvadis data.

Thank you for the suggestion. We added information about the minimum correlation coefficient for both Geuvadis and GTEx data in the Results section:

“At 1% FDR (corresponding to a minimum correlation coefficient of 0.16), we found 9,384 significantly co-expressed gene pairs (COPs) within 1MB of each other (8,716 correlated positively and 668 correlated negatively), which correspond to 9,030 distinct genes.”

“We identified 64,320 distinct COPs across tissues (FDR 1%, corresponding to a minimum correlation between 0.12 and 0.25, depending on the tissue).”

3. Regarding the negative COPs, the authors concluded "Overall, we find negative COPs to be much less numerous than positive COPs, and their lower tissue-conservation suggests a less important role for this category of COPs." I think, this is wrong. In contrast, it may

indicate negative COPs are more tissue-specific, as their results clearly showed. Tissue-specific regulation is no less important than universal regulation. The sentence "This indicates that negative COPs may be less relevant across human tissues." has the same problem.

We thank the reviewer for the feedback in interpreting these results. The text now reads "Overall, we find negative COPs to be less numerous than positive COPs **and display higher tissue-specificity.**"

Moreover, we removed the sentence "This indicates that negative COPs may be less relevant across human tissues"

Reviewers' Comments:

Reviewer #2:

Remarks to the Author:

All my concerns were properly addressed.